# Contribution of behavioural variability to representational drift

**Sadra Sadeh\*, Claudia Clopath\***

Department of Bioengineering, Imperial College London, London, United Kingdom

**Abstract** Neuronal responses to similar stimuli change dynamically over time, raising the question of how internal representations can provide a stable substrate for neural coding. Recent work has suggested a large degree of drift in neural representations even in sensory cortices, which are believed to store stable representations of the external world. While the drift of these representations is mostly characterized in relation to external stimuli, the behavioural state of the animal (for instance, the level of arousal) is also known to strongly modulate the neural activity. We therefore asked how the variability of such modulatory mechanisms can contribute to representational changes. We analysed large-scale recording of neural activity from the Allen Brain Observatory, which was used before to document representational drift in the mouse visual cortex. We found that, within these datasets, behavioural variability significantly contributes to representational changes. This effect was broadcasted across various cortical areas in the mouse, including the primary visual cortex, higher order visual areas, and even regions not primarily linked to vision like hippocampus. Our computational modelling suggests that these results are consistent with independent modulation of neural activity by behaviour over slower timescales. Importantly, our analysis suggests that reliable but variable modulation of neural representations by behaviour can be misinterpreted as representational drift if neuronal representations are only characterized in the stimulus space and marginalized over behavioural parameters.

**\*For correspondence:**
s.sadeh@imperial.ac.uk (SS);
c.clopath@imperial.ac.uk (CC)

**Competing interest:** The authors declare that no competing interests exist.

## Editor's evaluation

This work builds on rapidly accumulating evidence for the importance of measuring and accounting for behaviour in neural data, and will be of interest to a broad neuroscience audience. Analyses of Allen Brain Atlas datasets show that sensory representations change and match up reliably with behavioural state. The article's main conclusions are supported by the data and analyses, and the work raises important questions about previous accounts of the sources of representational drift in sensory areas of the brain.

## Introduction

Neuronal responses to stimuli, contexts, or tasks change over time, creating a drift of representations from their original patterns (*Deitch et al., 2021*; *Driscoll et al., 2017*; *Lütcke et al., 2013*; *Marks and Goard, 2021*; *Schoonover et al., 2021*; *Ziv et al., 2013*). This representational drift can reflect the presence of intrinsic noise or plasticity in the circuitry and, depending on its origin, can be detrimental to or beneficial for the neural code (*Clopath et al., 2017*; *Rule et al., 2019*). Understanding the mechanisms contributing to the emergence of representational drift can therefore shed light on its relevance for neural computation (*Lütcke et al., 2013*; *Rule et al., 2019*).

Representational drift can arise from a variety of sources, including bottom-up mechanisms, like changes in the feedforward input to neurons or from a dynamic reorganization of recurrent interactions in the network. Another important source of variability that can contribute to representational drift is changes in the behavioural state of the animal. Spontaneous behaviour has in fact been shown

to heavily modulate responses in awake behaving animals (*Musall et al., 2019*; *Niell and Stryker, 2010*; *Stringer et al., 2019*). Drift of behavioural state – for example, gradual changes in attention, arousal, or running – can therefore change the way neural activity is modulated by top-down mechanisms (*Niell and Stryker, 2010*; *Vinck et al., 2015*) over different timescales.

The exact manner in which such top-down mechanisms modulate the neural activity (*Cohen-Kashi Malina et al., 2021*; *Dipoppa et al., 2018*; *Fu et al., 2014*; *Garcia Del Molino et al., 2017*; *Pakan et al., 2016*) would in turn determine how behavioural variability affects the representational drift. One possibility is that stimulus-evoked responses are just scaled by arousal or running, as suggested by gain models (*Ferguson and Cardin, 2020*). Under this scenario, the behavioural state of the animal can modulate the similarity of sensory representations across multiple repeats of the same stimulus (representational similarity) by increasing or decreasing the signal-to-noise ratio. Another possibility is that the behaviour contributes independently to neuronal activity, and hence representational similarity is better described in a parameter space where internal and external parameters conjointly define the neural code. Under the latter scenario, variability in behavioural 'signal' could be perceived as noise from the viewpoint of sensory representations, and could therefore be potentially mistaken as representational drift.

To delineate the contribution of behavioural variability to representational drift and shed light on the involved mechanisms, we analysed publicly available datasets from the Allen Brain Observatory (*de Vries et al., 2020*; *Siegle et al., 2021*). These datasets provide a good opportunity to systematically address this question as standardized visual stimulation and large-scale recordings of population activity in response to multiple repeats of the same stimuli are combined with systematic measurements of behavioural parameters (like running speed and pupil size) across a large number of animals. Our analysis suggested that changes in the behavioural state of the animal can strongly modulate the similarity of neuronal representations in response to multiple repeats of the same stimulus. In fact, within the datasets analysed, a significant fraction of what may be described as representational drift in a sensory cortex can be attributed to behavioural variability. Our results suggest that the contribution of behavioural variability to changes in neuronal activity should be carefully quantified and considered in the analysis of representational similarity and representational drift.

## Results

### Representational similarity depends on the behavioural state of the animal

We analysed publicly available, large-scale, standardized in vivo physiology datasets recently published by the Allen Brain Observatory (*Siegle et al., 2021*). The electrophysiology datasets obtained via Neuropixels probes (*Jun et al., 2017*) provide the possibility of studying the spiking activity of a large number of units to visual stimuli (see 'Methods'). We studied similarity of neural activity in response to multiple repeats of the same natural movie (*Figure 1a*).

Previous studies have reported significant changes in stimulus-evoked representations even in sensory cortices, over the timescales of multiple hours to multiple days and weeks (*Deitch et al., 2021*; *Marks and Goard, 2021*; *Schoonover et al., 2021*). The Neuropixels electrophysiology datasets provide the opportunity of studying these representational changes while accounting for behavioural changes, although over a faster timescale (hours). Similar representational drift has in fact been reported for another dataset (obtained via two-photon calcium imaging) over the course of multiple days (*Deitch et al., 2021*). The insights obtained from this analysis may therefore help in understanding the mechanisms underlying representational drift over longer timescales.

To shed light on the involved mechanisms, we contrasted two potential scenarios (*Figure 1b*). Changes in population activity in response to the same stimulus can arise from a completely random and independently added noise. Alternatively, modulation of activity by other (stimulus-independent) factors like behavioural modulation can also contribute to these changes (*Figure 1b*). To delineate between these two scenarios, we characterized how neuronal representations change across repetitions of the same stimulus (*Figure 1a and b*). This was quantified by a measure of representational similarity (RS), which was characterized as the similarity of responses, at the population level, to multiple repeats of the stimulus (see 'Methods' and *Figure 1—figure supplement 1*). Representational drift

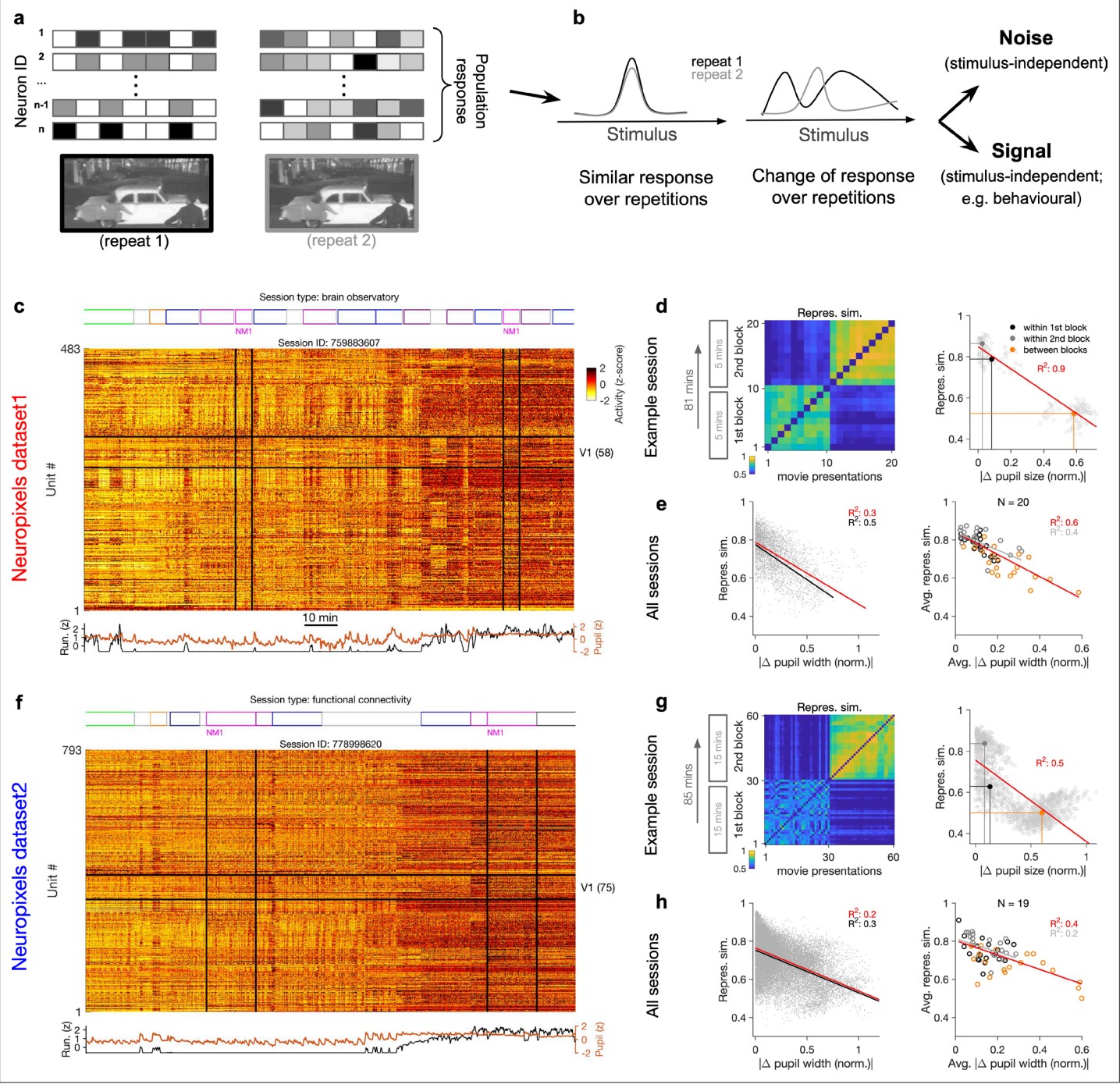

**Figure 1.** Representational similarity depends on the behavioural state of the animal. (**a**) Illustration of the response of a population of neurons to a stimulus (e.g. a natural movie) which is shown twice (left versus right). (**b**) Neuronal response to different repetitions of the same stimulus can remain the same (left) or change from the original pattern (right), leading to a change in neural representations. This change can arise from an added stimulus-independent noise, which is randomly and independently changing over each repetition. Alternatively, it can be a result of another signal, which is modulated according to a stimulus-independent parameter (e.g. behaviour) that is changing over each repetition. (**c**) Population activity composed of activity of 483 units in an example recording session in response to different stimuli (timing of different stimulus types denoted on the top). Spiking activity of units is averaged in bins of 1 s and z-scored across the entire session for each unit. Units in primary visual cortex (V1; 58 units) and the two blocks of presentation of natural movie 1 (NM1) are highlighted by the black lines. Bottom: pupil size and running speed of the animal (z-scored). (**d**) Representational similarity between different presentations of NM1. It is calculated as the correlation coefficient of vectors of population response of V1 units to movie repeats (see 'Methods'). Left: the matrix of representational similarity for all pairs of movie repeats within and across the two blocks of presentation. Right: representational similarity as a function of the pupil change, which is quantified as the normalized absolute difference of

*Figure 1 continued on next page*

*Figure 1 continued*

the average pupil size during presentations (see 'Methods'). The best-fitted regression line (using least-squares method) and the $R^2$ value are shown. Filled circles show the average values within and between blocks. (**e**) Same as (**d**) right for all recording sessions. Left: data similar to (**d**) grey dots are concatenated across all mice and the best-fitted regression line to the whole data is plotted. Black line shows the fit when movie repeats with significant change in the average running speed of the animal is considered (80th percentile). Right: the average values within and between blocks filled circles in (**d**) are plotted for all mice and the fitted regression line to these average values is plotted. Grey lines and $R^2$ values indicate the fit to within-block data only. N: number of mice. (**f–h**) Same as (**c–e**) for a different dataset. Source data (for normalized changes in pupil width and representational similarity between pairs of movie repeats) are provided for individual sessions across the two datasets (*Figure 1—source data 1*).

The online version of this article includes the following source data and figure supplement(s) for figure 1:

**Source data 1.** Related to *Figure 1*.

**Figure supplement 1.** Characterization and quantification of representational similarity (RS) and representational drift.

**Figure supplement 2.** Relation between behavioural changes and representational similarity in other sessions and for all units.

**Figure supplement 3.** Dependence of representational similarity on behavioural change when calculated from z-scored activity and from activity rendered in longer time bins.

**Figure supplement 4.** Dependence of representational similarity on behavioural change in wild-type (WT) mice and for male/female animals separately.

**Figure supplement 5.** Representational drift between the two blocks of stimulus presentation in animals with different levels of behavioural variability.

was calculated as the difference of RS within and between multiple repeats of the natural movies (see 'Methods').

Metrics of representational drift (e.g. representational drift index [RDI]; *Marks and Goard, 2021*; see 'Methods') are often calculated from RS. We therefore based our analysis primarily on comparing RS and its relationship with sensory and behavioural parameters as this contained more information. RDI was calculated as a single metric to quantify representational drift between blocks of presentation in each session. The drop in representational similarity between individual pairs of movie repeats was used to analyse representational drift on a finer scale (this drop is mathematically related to representational drift; see 'Methods'). The main behavioural parameters we analysed were the size of pupil and the running speed of animals. We refer to more slowly changing dynamics of these parameters (compared to sensory-evoked responses) as the behavioural state of the animal, and changes in these states are described as behavioural variability.

Our analysis was performed in two datasets with different structure of stimulus presentations (*Figure 1c and f*; see *Supplementary file 1*). In each dataset, the natural movie (30 s long) is presented multiple times in each block of presentation (10 and 30 repeats for dataset 1 and dataset 2, respectively). We analysed the data for two blocks of presentation separated by more than an hour (*Figure 1c and f*). For each presentation of the natural movie, we calculated a population vector of responses from the average activity of all the units recorded in the primary visual cortex (V1), in bin widths of 1 s starting from the onset of movie presentation ('Methods'). Representational similarity between two repeats of the natural movie was quantified by the correlation coefficient of the population vectors (see *Figure 1—figure supplement 1c* and 'Methods').

Previous analysis has shown that representational similarity (as defined above) is higher within a block of presentation and decreases significantly between different blocks, both in Neuropixels and calcium imaging datasets (*Deitch et al., 2021*). Our results confirmed this observation, but we also found that representational similarity is strongly modulated by the behavioural state of the animal. This was most visible in sessions where the behavioural state (as assayed by pupil diameter and the average running speed) changed clearly between the two repeats of the movie (*Figure 1c and f*). We observed that, firstly, change in the behavioural state strongly reduced the representational similarity between the two blocks (*Figure 1d and g*), reminiscent of the representational drift which has been reported over the scale of hours to days (*Deitch et al., 2021*; *Marks and Goard, 2021*; *Schoonover et al., 2021*). Secondly, increased pupil diameter and running during the second block of presentation in fact increased the similarity of responses to the same movie within that block (*Figure 1d and g*, left). Overall, there was a significant drop of representational similarity between the movie repeats in which the animal experienced the most changes in the average pupil size (*Figure 1d and g*, right). These results indicate that the behavioural state of the animal can bidirectionally modulate the representational similarity across repeats of the same stimulus.

We found similar dependence of representational similarity on the pupil change for other sessions (*Figure 1—figure supplement 2a*) and across all animals (*Figure 1e and h*). The effect was more prominent when focusing on movie repeats with significant changes in the average running (*Figure 1e and h*, left, black lines). Similar trend was also observed when considering units from all recorded regions, instead of only focusing on V1 units (*Figure 1—figure supplement 2b*). We also observed the same trend when repeating the analysis within blocks (*Figure 1d–h*, right, grey lines, and *Figure 1—figure supplement 2b*), although the drop of representational similarity across blocks was more prominent due to more drastic behavioural changes between the blocks, which is expected from the slow timescale of changes in behavioural states.

In the above analysis, we considered the actual spiking activity of the units to build the population vectors. Calculating the representational similarity from these vectors can potentially bias the estimate by increasing the impact of highly active neurons. For instance, if the units which are firing higher remain consistently active, they may lead to some similarity of population vectors even independent of stimulus-evoked responses. To control for variance in the average activity of units, we repeated our analysis for population vectors composed of z-scored responses (as shown in *Figure 1c and f*; see 'Methods'). Overall, representational similarity diminished when calculated from the z-scored activity (*Figure 1—figure supplement 3a*). However, we observed the same trend in terms of dependence on the behavioural state, whereby larger changes in pupil size were correlated with larger changes in representational similarity (*Figure 1—figure supplement 3a*).

We performed our original analysis with vectors of activity rendered in time bins of 1 s, corresponding to the interval of presentation of each frame of the natural movie ('Methods'). We tested whether our results hold for responses rendered in different time bins by testing longer time bins (2 s; *Figure 1—figure supplement 3b*). Overall, representational similarity was higher as expected from averaging the activity. However, we observed similar drop of representational similarity with increases in pupil size (*Figure 1—figure supplement 3b*).

Our previous analyses were performed on wild-type mice as well as mice from three different transgenic lines (Pvalb-IRES-Cre × Ai32, n = 8; Sst-IRES-Cre × Ai32, n = 12; and Vip-IRES-Cre × Ai32, n = 8; see *Supplementary file 1*; *Siegle et al., 2021*). To control for possible differences between different strains, we repeated our analysis for recording sessions in wild-type mice only and observed similar results (*Figure 1—figure supplement 4a*). Our results also held when analysing female and male animals separately (female mice comprised a smaller fraction of the datasets; ~20%) (*Figure 1—figure supplement 4b*).

In our results above, we observed large changes in representational similarity between blocks of stimulus presentation with strong behavioural changes. But this effect can be confounded by the passage of time between the blocks, which may lead to other sources of variability such as changes in the excitability of neurons for instance. To account for this, we analysed the average block-wise representational similarity in individual animals (*Figure 1—figure supplement 5*). For each animal, we quantified the RDI between the two blocks as the normalized change of average correlations within and across blocks (*Marks and Goard, 2021*; see 'Methods').

If the passage of time contributes strongly to the drop in representational similarity between the two blocks, we should see comparable levels of representational drift across animals with different levels of behavioural variability. Conversely, if the change in behaviour is the main factor contributing to the drop in representational similarity between the two blocks, we should see stronger levels of representational drifts for animals with larger behavioural variability. We indeed found evidence in favour of the latter: representational similarity remained rather stable for those animals which did not show large behavioural changes between the two blocks (*Figure 1—figure supplement 5a*). That is, passage of time per se did not contribute strongly to representational drift. Largest representational drifts were observed for animals with the largest changes in the average pupil width between the two blocks (*Figure 1—figure supplement 5a*). In fact, there was a good correlation between the two across animals (*Figure 1—figure supplement 5b*).

The relationship was weaker in Neuropixels dataset 2, whereby more repeats of the natural movie are shown in each block (60 repeats for the total of 30 min versus 20 repeats for 10 min in Neuropixels dataset 1). Longer blocks of stimulus presentation increase the chance of behavioural changes within the blocks, which can in turn make the average block-wise representational similarity a less reliable metric. In line with this reasoning, further scrutiny into an outlier (with small average pupil changes but

rather large representational drift between the two blocks; *Figure 1—figure supplement 5b*, right) revealed that changes in the running speed of the animal within each block can be the factor contributing to changes in representational similarity (*Figure 1—figure supplement 5c*).

Taken together, these results suggest that, in awake behaving animals, variability of the behavioural state of the animal can be an important factor contributing to the modulation of representational similarity.

## Evidence for independent modulation of responses by stimulus and behaviour

What is the mechanism by which behavioural state modulates representational similarity? Changes in pupil area are correlated with the level of arousal (*Bradley et al., 2008*), which can modulate the neuronal gain (*Cohen-Kashi Malina et al., 2021*). We therefore studied a possible gain model in which changes in pupil size modulate the neuronal responses to sensory inputs (see *Figure 2a* and 'Methods'). Alternatively, rather than scaling the stimulus-induced signal, behaviour can contribute independently to neuronal activity (*Dipoppa et al., 2018*; *Saleem et al., 2013*). We therefore compared the gain model to a model in which the neural tuning was obtained by an independent mixing of stimulus and behavioural signals (see *Figure 2b* and 'Methods').

In each model, we calculated representational similarity from the neuronal responses to presentations of the same stimulus and plotted that against the relative behavioural parameter (B) across the repeats ($B_i/B_j$ , for the $i$th and $j$th repeats) (*Figure 2c and d*). Both models showed, on average, a similar dependence of representational similarity on relative behaviour (*Figure 2c and d*; the gain model only showed the same pattern if the signal was scaled by the behaviour; we observed different patterns, if behaviour scaled the noise, or both the signal and the noise; *Figure 2—figure supplement 1a and b*).

To compare different models with the experimental results, we took the relative pupil size as a proxy for relative behaviour and plotted the representational similarity of all units against it (*Figure 2e*). This revealed a similar average dependence as the signal-gain model and the independent-mixing model (*Figure 2c–e*). We observed a similar dependence for both datasets, and for most individual recording sessions within each dataset (*Figure 2—figure supplement 1c–f*). Similar results were observed when representational similarity was calculated from V1 units or all recorded units (*Figure 2—figure supplement 1c–f*).

We then asked how, at the level of individual units, the modulations of responses by stimulus and behaviour are related to each other (*Figure 2f and g*). To this end, instead of calculating representational similarity at the population level, we quantified the similarity of individual units' responses to multiple repeats of the stimulus (stimulus reliability; see 'Methods' and *Figure 1—figure supplement 1d*). In the signal-gain model, stimulus reliability was highly correlated with behavioural modulation of units (*Figure 2f*). This is a consequence of the scaling of the signal by the behaviour, which implies that neurons with higher signal component also show higher modulation with the behavioural parameter (see 'Methods'). The signal-gain model therefore predicts that neurons which are strongly modulated by the stimulus also show strong modulation by the behaviour (*Figure 2f*). In contrast, the independent-mixing model predicted an independent relationship between stimulus and behavioural modulation of individual units (*Figure 2g*).

We tested these predictions in experimental data by calculating behavioural modulation and stimulus reliability of individual units in all mice across both datasets. Behavioural modulation was calculated as the correlation of each unit's activity with pupil size, and stimulus reliability was obtained as the average correlation of each unit's activity vectors across multiple repeats of the natural movie (see 'Methods' and *Figure 1—figure supplement 1d*). As opposed to the signal-gain model, we did not observe a correlation between stimulus and behavioural modulation (*Figure 2h*). In fact, a regression analysis suggested that the two modulations are independent of each other in both datasets (*Figure 2h*), consistent with the independent-mixing model. The marginal distributions matched better with experimental distributions when we increased noise (x1.5 N) or decreased the behavioural signal (x0.5 B) (*Figure 2—figure supplement 1g*).

Overall, there was a wide distribution of stimulus reliability (*Figure 2—figure supplement 2a*) and behavioural modulation (*Figure 2—figure supplement 2c*) across recorded units, with patterns highly consistent across the two datasets. Most V1 units showed variable responses to repeats of the natural

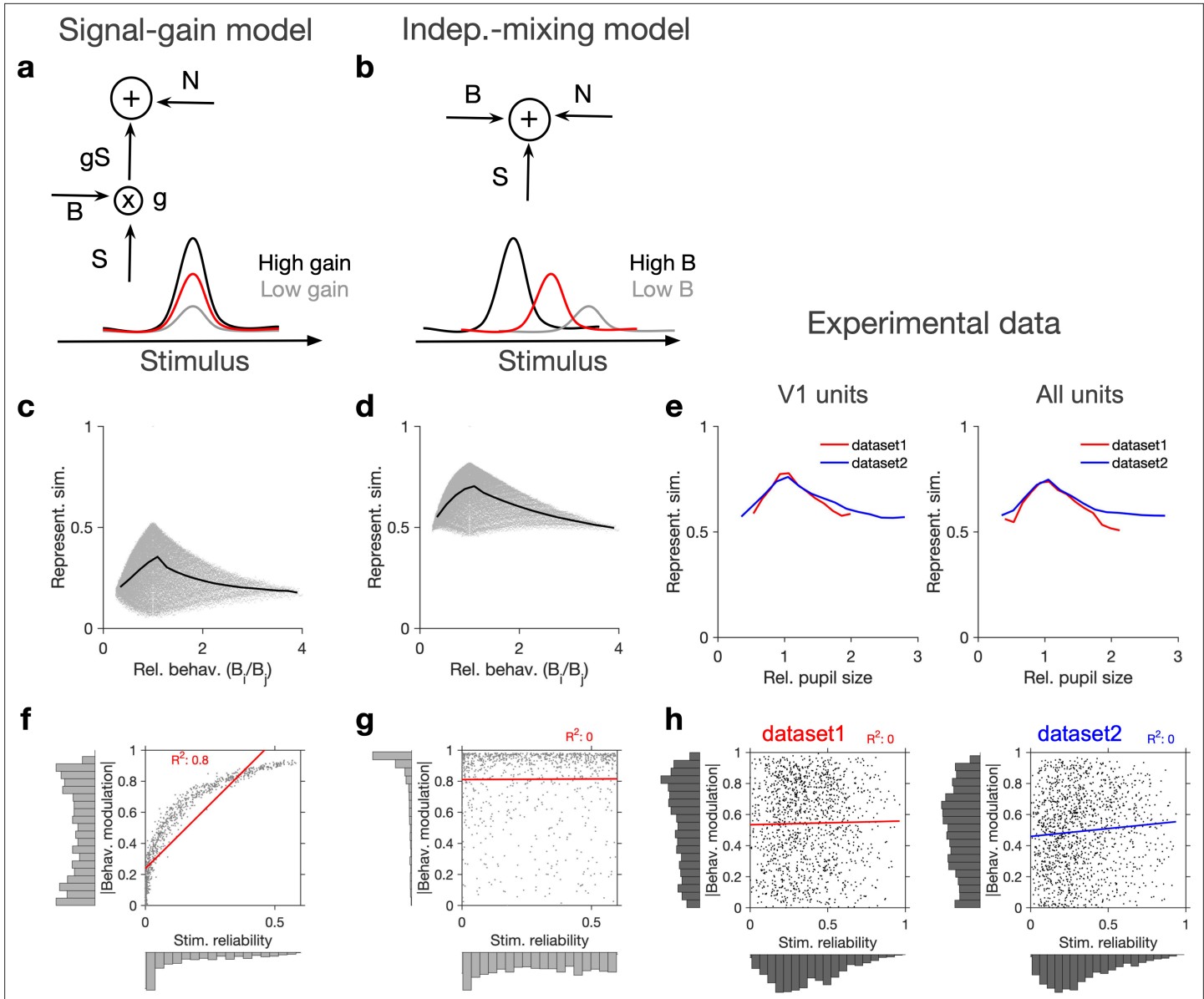

**Figure 2.** Independent modulation of activity by stimulus and behaviour. (**a**) Schematic of a signal-gain model in which behaviour controls the gain with which the stimulus-driven signal is scaled. Individual units are driven differently with the stimulus, leading to different tuning curves which determines their stimulus signal, $S$. Behavioural parameter, B, sets the gain, $g$, with which the stimulus signal is scaled, before being combined with the noise term, N, to give rise to the final response. S is the same across repetitions of the stimulus, while N is changing on every repeat (see *Figure 2—figure supplement 1a* and 'Methods'). (**b**) An alternative model (independent-mixing) in which the response of a unit is determined by the summation of its independent tuning to stimulus, S (red) and behaviour, B (black: high B; grey: low B), combined with noise, N (see 'Methods' for details). (**c**) Representational similarity of population responses to different repeats of the stimulus as a function of the relative behavioural parameter ($B_i/B_j$) in the signal-gain model. Black line shows the average (in 20 bins). (**d**) Same as (**c**), for the independent-mixing model. (**e**) Same as (**c, d**) for the experimental data from Neuropixels dataset 1 (red) or dataset 2 (blue). For each pair of movie repeats, the average representational similarity of population responses (left: V1 units; right: all units) are plotted against the relative pupil size ($P_i/P_j$). (**f, g**) Relation between behavioural modulation and stimulus reliability of units in different models. The stimulus signal-gain model predicts a strong dependence between behavioural modulation and stimulus reliability of units (**f**), whereas the independent-mixing model predicts no correlation between the two (**g**). Best-fitted regression lines and $R^2$ values are shown for each case. Marginal distributions are shown on each axis. (**h**) Data from V1 units in the two datasets show no relationship between the stimulus reliability of units and their absolute behavioural modulation, as quantified by the best-fitted regression lines and $R^2$ values. Stimulus reliability is computed as the average correlation coefficient of each unit's activity vector across repetitions of the natural movie, and behavioural modulation is calculated as the correlation coefficient of each unit's activity with the pupil size (see 'Methods'). Marginal distributions are shown. Source data (for stimulus reliability of V1 units and their modulation by pupil size) are provided for individual sessions across the two datasets (*Figure 2—source data 1*).

*Figure 2 continued on next page*

movie, as indicated by the peak of the distribution at low stimulus reliability (*Figure 2—figure supplement 2a*). However, the distribution had a long tail composed of units with high stimulus reliability, which showed highly reliable responses across repeats of the movie (*Figure 2—figure supplement 2a and b*). There was a wide spectrum of behavioural modulation too, with most units showing positive correlations with pupil size (*Figure 2—figure supplement 2c and d*), and a smaller population of negatively modulated units (*Figure 2—figure supplement 2c*).

The units that showed significant modulation with the stimulus were not necessarily modulated strongly by the behaviour, and vice versa; in fact, it was possible to find example units from all four combinations of weak/strong × stimulus/behavioural modulations (*Figure 2—figure supplement 2e and f*). A clear example of the segregation of stimulus and behavioural modulation was observed in CA1, where the units showed, on average, very weak stimulus reliability across movie repeats, consistently across different mice and datasets (*Figure 2—figure supplement 3a*). However, they were largely modulated by behaviour, to an extent comparable to V1 units (*Figure 2—figure supplement*

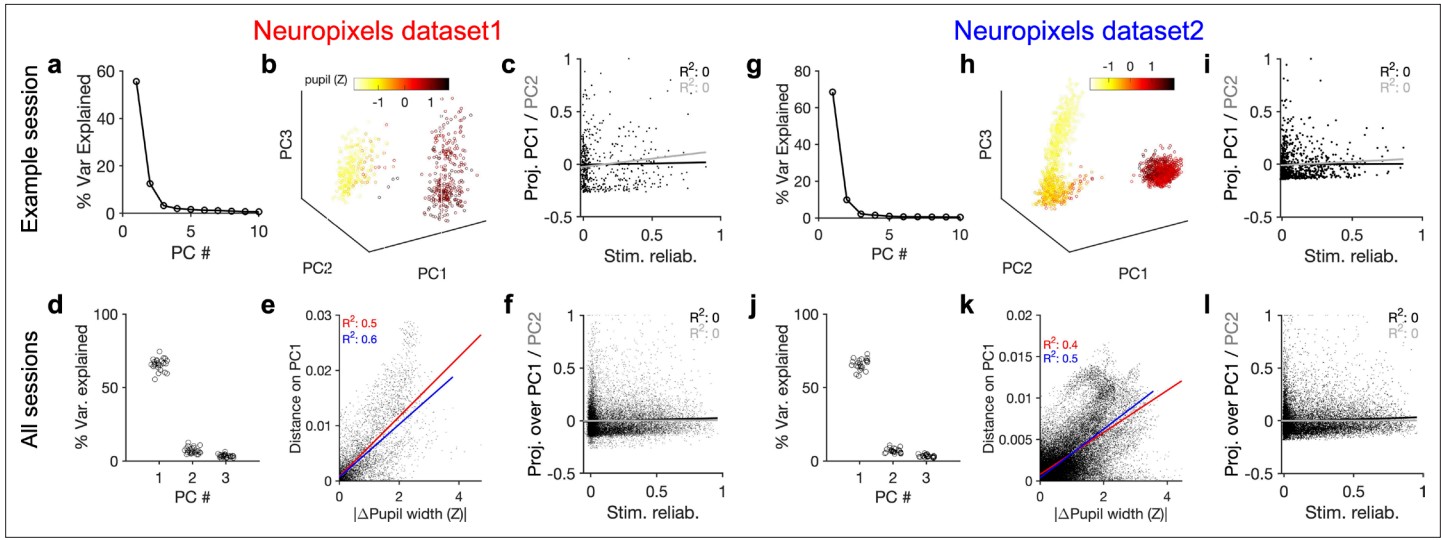

**Figure 3.** Behavioural variability modulates the low-dimensional components of population activity independent of stimulus reliability. (**a**) Relative contribution of the first 10 principal components (PCs) of population responses to the variability of activity (quantified by the fraction of explained variance by each PC) for an example session (same as shown in *Figure 1*). (**b**) Population activity in the low-dimensional space of the first three PCs (see 'Methods' for details). Pseudo colour code shows the pupil size at different times, indicating that the sudden transition in the low-dimensional space of activity is correlated with changes in the behavioural state. (**c**) Projection of units' activity over PC1 (black) or PC2 (grey) (respective PC loadings) versus stimulus reliability of the units. It reveals no correlation between the two, as quantified by best-fitted regression lines in each case (the best-fitted regression lines and $R^2$ values shown by respective colours). PC loadings are normalized to the maximum value in each case. (**d**) Fraction of variance explained by PCs 1–3 for all sessions. (**e**) Distance on PC1 versus the difference of the average pupil size between different time points on the trajectory for all sessions. For each pair of non-identical time points on the PC trajectory, the absolute difference of PC1 components is calculated as the distance on PC and plotted against the absolute difference of the average pupil width (normalized by z-scoring across the entire session). Best-fitted regression line and the corresponding $R^2$ value are shown in red for all data points, and in blue for data points with small time difference between the pairs (see *Figure 3—figure supplement 2*). (**f**) Same as (**c**) for all units from all sessions in dataset 1. (**g–l**) Same as (**a–f**) for Neuropixels dataset 2.

The online version of this article includes the following figure supplement(s) for figure 3:

**Figure supplement 1.** Relation of the principal components (PCs) of neural activity to the average activity of units and their stimulus reliability.

**Figure supplement 2.** Principal component (PC)1 and its relation to difference in pupil size and time.

*3a–c*). Taken together, these results suggest that, rather than scaling the stimulus-evoked responses, behaviour modulates the activity in a more independent and heterogeneous manner.

## Behavioural variability modulates the low-dimensional components of population activity independent of stimulus reliability

If the behavioural state of the animal modulates the neuronal responses independently of the stimulus, it should be possible to see its signature in the low-dimensional space of neural activity. To test this, we analysed the principal components (PCs) of population responses in individual sessions (*Figure 3*; see 'Methods'). For the two example sessions we analysed previously (shown in *Figure 1c and f*), the first two PCs explained a significant fraction of variance (*Figure 3a and g*). Low-dimensional population activity showed a distinct transition between two behavioural states, which were corresponding to low versus high arousal, as quantified by different pupil sizes (*Figure 3b and h*). The first PC, which explained most of the variance, was strongly correlated with both pupil size and running speed (*Figure 3—figure supplement 1a*). These results suggest that behavioural modulation contributes significantly to the low-dimensional variability of neural activity.

To link the low-dimensional modulation of activity by behaviour to single neurons, we next analysed the projection of units' activity over the PCs by looking at PC loadings. Individual units were projected in the PC space according to their respective PC loadings (PC projections). Visualizing the average activity of units in the space of PC projections suggested a spectrum of weakly to highly active units (*Figure 3—figure supplement 1b*). In fact, neural projections over the first two PCs were correlated with the average activity of neurons (*Figure 3—figure supplement 1c*). In contrast to the average activity, the PC projections did not reveal any relationship with the stimulus reliability of units (*Figure 3c and i*), suggesting that the low-dimensional neural activity is modulated independently of stimulus-evoked modulation of responses.

These results were remarkably consistent across different datasets and across difference mice. The first two PCs explained similar levels of variance across more than 20 mice in each dataset (*Figure 3d and j*). The distance over PC1 was highly correlated with the difference of pupil size on the trajectory (*Figure 3e and k*). Importantly, this relationship also held when controlled for the passage of time (*Figure 3e and k*, *Figure 3—figure supplement 2*). In both datasets, the regression analysis revealed no relationship between the two PC projections and the stimulus reliability of units (*Figure 3f and l*; see *Figure 3—figure supplement 1d* for individual sessions). We therefore conclude that behaviour significantly modulates the low-dimensional components of neural activity, but this modulation does not specifically enhance the activity of neurons which are more reliably representing the stimulus.

## Behaviour modulates the setpoint of responses

To gain further insight into how the behaviour modulates the low-dimensional pattern of population activity, we analysed the relation between behavioural parameters and the average activity of units. In the two example sessions analysed previously (shown in *Figure 1*), there was a transition in the average pupil size and running speed in the second block, which was correlated with an overall increase in the average population activity (*Figure 4a and e*, *Figure 4—figure supplement 1a*). In general, change in the pupil size explained a significant fraction of changes in population activity of V1 units in all sessions (*Figure 4—figure supplement 1b*).

We also looked at the average activity of individual units across all movie frames and repetitions (their setpoints). Units had a wide range of setpoints, which were relatively stable within each block (small variance across repetitions relative to the mean) (*Figure 4—figure supplement 1c*). However, the setpoints changed upon transition to the next block, with most units increasing their setpoints, without an apparent link to their previous setpoint levels (*Figure 4—figure supplement 1d*). The population vectors composed of setpoints in each repeat can be further used to quantify setpoint similarity (*Figure 4b and f*). Within-block correlations were high, indicating the stability of setpoints when behavioural changes were minimum – although occasional, minor changes of pupil size still modulated these correlations (*Figure 4b and f*). Most changes in setpoint similarity, however, appeared between the blocks, when the animal experienced the largest change in its behavioural state.

Quantifying the dependence of setpoint similarity on changes in pupil size revealed a strong relationship, both for V1 units and for all recorded units (*Figure 4c and g*). The relationship was rather stable when calculated from responses to single frames of movie presentation, instead of the average

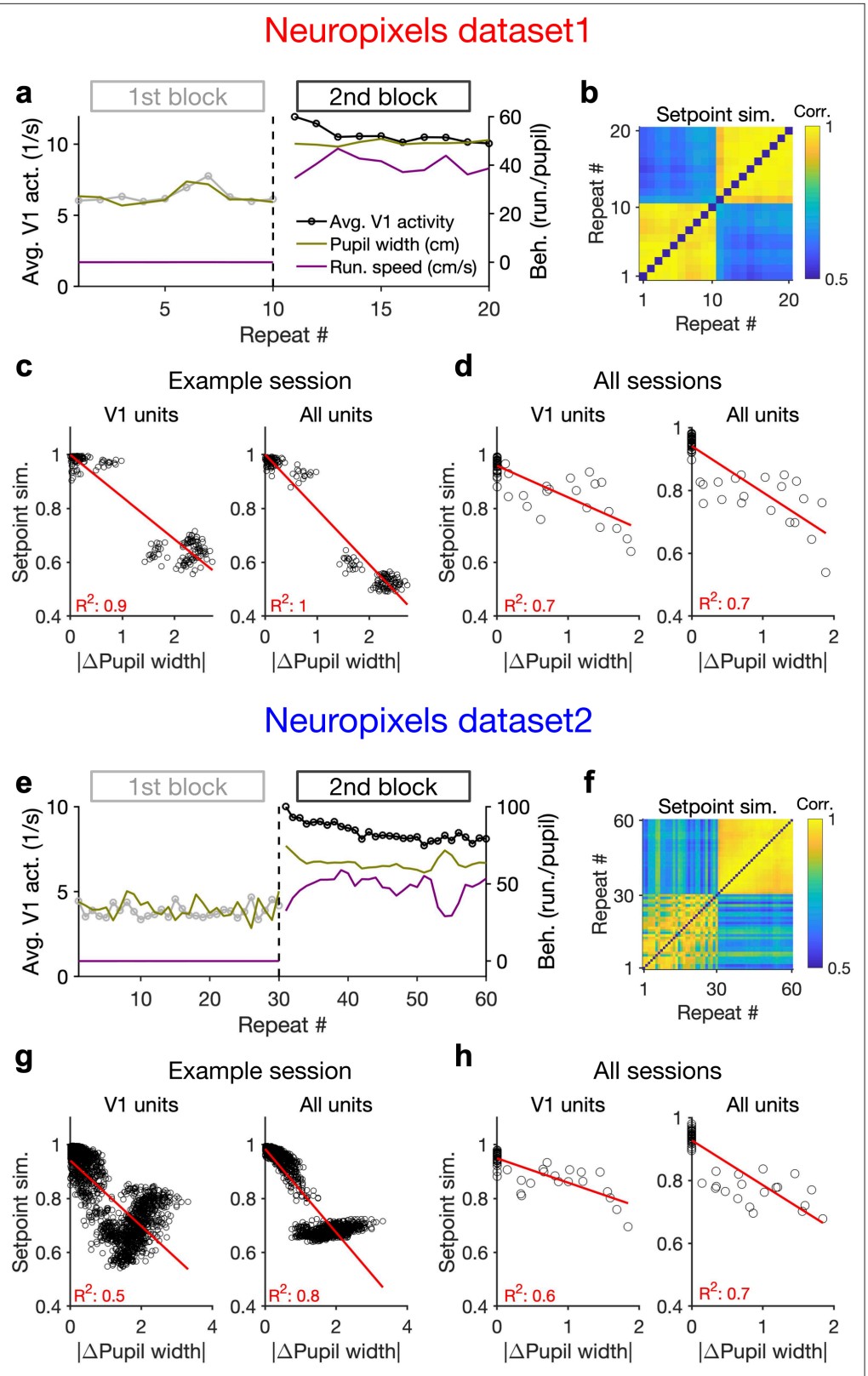

**Figure 4.** Behaviour modulates the setpoint of responses. (**a**), Average population activity and behavioural parameters (pupil size and running speed) during the first and second blocks of presentation of natural movie 1 (same examples sessions as *Figure 1*). Grey, first block; black, second block; each point corresponding to the average in one repeat of the movie. (**b**) Setpoint similarity is calculated as the correlation coefficient of

*Figure 4 continued on next page*

*Figure 4 continued*

population vectors composed of average activity of units during each repeat of movie presentation. Change in the behavioural state (as quantified by the pupil size) between the two blocks is correlated with a drastic decrease in the average between-block setpoint similarity. Note that transient changes of pupil within each block also modulate the setpoint similarity. (**c**), Setpoint similarity (as in **b**) as a function of change in pupil size (z-scored pupil width) between the movie repeats, when the population vectors of setpoints are composed of V1 units (left) or all recorded units (right). (**d**), Dependence of setpoint similarity on pupil change for all sessions, calculated from within-block and across-block averages in each session. (**e–h**), Same as (**a–d**) for dataset 2. Source data (for the change in pupil width and setpoint similarity between pairs of movie repeats) are provided for individual sessions across the two datasets (*Figure 4—source data 1*).

The online version of this article includes the following source data and figure supplement(s) for figure 4:

**Source data 1.** Related to *Figure 4*.

**Figure supplement 1.** Average activity of units is modulated by behavioural state.

**Figure supplement 2.** Nonmonotonic relationship between setpoint similarity and behaviour.

**Figure supplement 3.** Changes in the pupil centre and their relation to setpoint similarity and changes in pupil width (for sessions in Neuropixels dataset 1).

**Figure supplement 4.** Changes in the pupil centre and their relation to setpoint similarity and changes in pupil width (for sessions in Neuropixels dataset 2).

**Figure supplement 5.** Overall average activity of units and average pupil width gradually increase during the recording session.

---

activity across the entire movie (*Figure 4—figure supplement 1e*). We obtained similar results when the dependence was calculated from the average block activity across all mice from both datasets (*Figure 4d and h*).

The relationship of setpoint similarity with changes in pupil size was not always monotonic. For the example session in *Figure 4g*, despite an overall drop of setpoint similarity with increased changes in pupil width, another trend was observed for large pupil changes. To understand what underlies this nonmonotonic trend we plotted setpoint similarity as a function of changes in both pupil size and running speed (*Figure 4—figure supplement 2*). The multidimensional space of changes in behaviour suggests that different trends exist at different regimes of behavioural changes (*Figure 4—figure supplement 2b*). Notably, limiting our analysis to the pairs of movie repeats with small changes in running speed revealed a monotonic relationship between setpoint similarity and changes in pupil size, with similar levels of regression between the two parameters (*Figure 4—figure supplement 2a–c*).

The opposite trend emerged at higher levels of changes in pupil size and running speed, and seemed to be related to changes in pupil size in the first block of presentation (*Figure 4—figure supplement 2d and e*). Indeed, movie presentations with higher (/lower) average pupil size in the first block showed smaller (/larger) setpoint similarity with the movie repeats in the second block, on average (*Figure 4—figure supplement 2d and e*). This trend was accompanied, and might be explained, by an average movement in the centre of the pupil between the two blocks (*Figure 4—figure supplement 2f*). Such a movement can shift the receptive fields, and therefore change the setpoint activity of units, leading to changes in setpoint similarity between the two blocks. This effect is expected to be higher for higher pupil sizes, and therefore the setpoint similarity is specifically reduced for movie presentations with larger pupil sizes in the first block.

The resulting change in the average visual input may also explain why the nonmonotonic trend is specifically present in V1 units and tends to disappear when all units are considered (*Figure 4g*). Consistent with this reasoning, shifts in the average centre of pupil also showed correlation with changes in setpoint similarity across many sessions (*Figure 4—figure supplements 3 and 4*). This was reminiscent of the dependence of setpoint similarity on pupil width and tended to be higher for sessions with strong correlation between changes in pupil size and position (*Figure 4—figure supplements 3 and 4*).

Our results therefore suggest that the changes in setpoint similarity can arise from complex interaction between multiple behavioural parameters and their modulation of neural activity. On a case-by-case basis, it remains to be seen how behavioural parameters and their interactions specifically modulate neural activity on finer scales. On average, however, there was a gradual increase of the

average pupil width during the recording session, which was paralleled by a gradual increase of the average activity of units (~50% increase for the average activity of all units across all mice in both datasets; *Figure 4—figure supplement 5*). The results of our analysis therefore suggest that the behavioural signal can modulate the setpoint of neural activity independent of and in addition to stimulus, and, in doing so, induce a similarity (/dissimilarity) of population responses when behaviour is stable (/changing).

Note that an unintuitive connotation of this finding is that quantifying response similarity from population vectors may reveal representational drift upon behavioural changes, even independent of stimulus-evoked modulation of activity. This is because the constancy of setpoint activity of units would lead to some degree of similarity between population vectors, even if the stimulus-evoked component is different (*Figure 2—figure supplement 3d*). The behaviourally induced component of similarity changes more slowly, leading to a drop in representational similarity on a longer timescale (e.g. between blocks of stimulus presentation, rather than within them). In line with this reasoning, we observed a similar drop of representational similarity in CA1 (*Figure 2—figure supplement 3e*), although individual units in this region had, on average, no reliable visual representations (*Figure 2—figure supplement 3a*). Modulation of the average setpoint activity – and hence setpoint similarity – by the behaviour can, therefore, contribute to representational similarity, in addition to specific stimulus-induced tuning of responses.

## Behaviour reliably modulates responses during active states

What distinguishes an independent behavioural signal from a modulatory component or from noise is that it brings about reliable responses for different states of behaviour. That is, there should exist

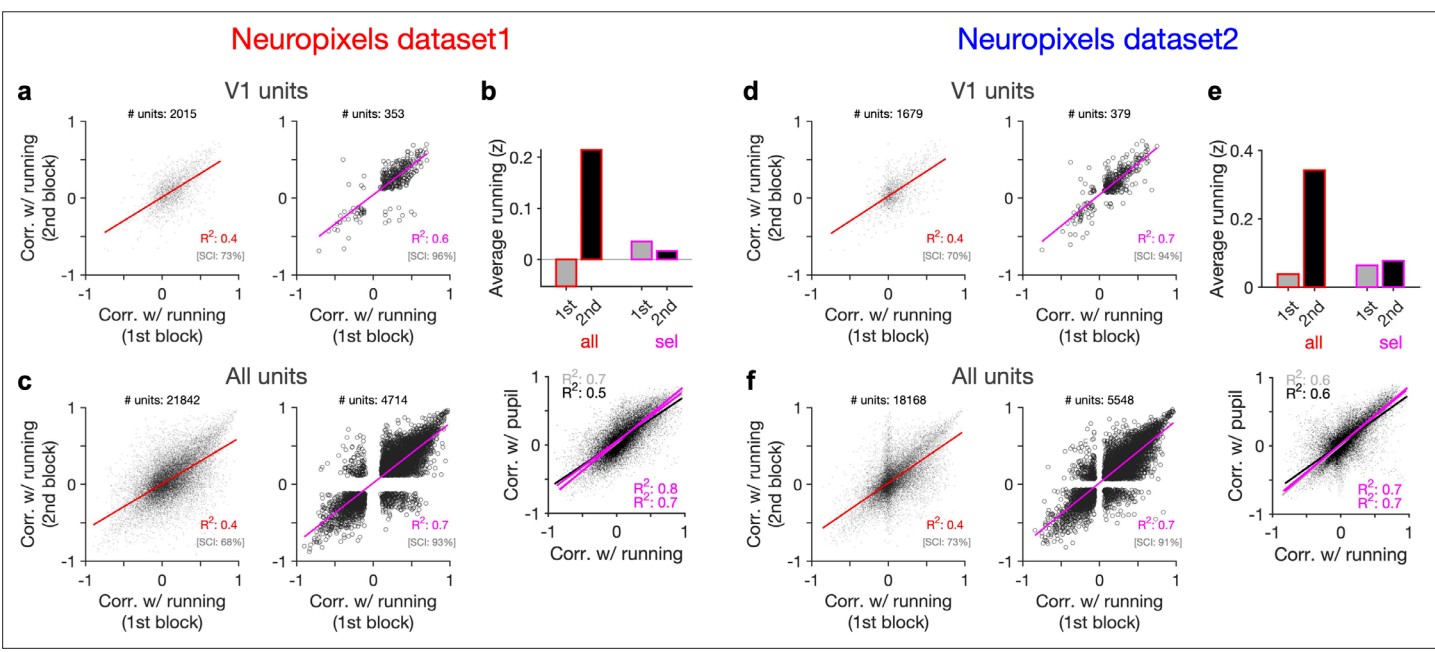

**Figure 5.** Behaviour reliably modulates responses during active states. (**a**) Correlation of activity with running during second block against the first block for all V1 units (left) and for selected sessions and units (right). In the latter case, only sessions with similar average running between the two blocks, and units with significant correlation with running, are selected (see *Figure 5—figure supplement 1* and 'Methods' for details). In addition to the regression analysis (quantified by the value of $R^2$), a second metric (sign constancy index [SCI]) is also used to quantify the reliability of modulations by behaviour. For each unit, the sign of correlation with running in different blocks is compared and the fraction of units with the same sign is reported as SCI. (**b**) Upper: average z-scored value of running in the first and second block across all units/sessions (all; red) and for selected ones (sel; magenta). Lower: correlation of all V1 units with pupil size against their correlation with running in first (grey) and second (black) blocks. Magenta: regression fits for selected units/sessions only. (**c**) Same as (**a**) for recorded units from all regions. (**d–f**) Same as (**a–c**) for dataset 2.

The online version of this article includes the following figure supplement(s) for figure 5:

**Figure supplement 1.** Significant modulation of units by behaviour.

**Figure supplement 2.** Consistent modulation of neuronal responses by behaviour across blocks of presentation of drifting gratings and across stimuli.

**Figure supplement 3.** Consistent modulation of neuronal responses by behaviour across stimuli, regions, and datasets.

a reliable and independent tuning of units with behavioural parameters (like pupil size or running speed). We therefore investigated more precisely how the neural activity is modulated by behaviour (*Figure 5*). We used the correlation of units' activity with running as a metric for behavioural tuning. To obtain a measure of significance of correlations, we calculated bootstrapped correlations (see 'Methods'). More than half of the units showed significant modulation by running, and the fraction and distribution of significant correlations were similar between the two blocks and across the two datasets (*Figure 5—figure supplement 1*).

Another way to assay the reliability of behavioural tuning is to test whether the correlation of units with behaviour remains stable between the two blocks of presentation (*Figure 5a and d*). Random correlations with running should be uncorrelated across the two repeats. In contrast, regression analysis revealed a good correlation between the two blocks (*Figure 5a and d*, left). The distributions of correlations with behaviour were also similar between the two blocks (*Figure 5—figure supplement 1*). Notably, focusing on sessions with similar levels of running between the two blocks (*Figure 5b and e*), and on units with significant behavioural modulation, improved the similarity of tuning between the two repeats (*Figure 5a and d*, right). Specifically, most units which were positively (/negatively) modulated during the first block remained positively (/negatively) modulated in the second block (*Figure 5a and d*, right). These results therefore suggest that a significant fraction of the population shows reliable modulation by running – similar result is expected for pupil, as we observed a high correlation between modulation of units with running and pupil in both datasets (*Figure 5b and e*, lower).

Our results held when repeating the analysis for all units instead of V1 units only (*Figure 5c and f*, *Figure 5—figure supplement 1*). We also observed similar results when quantifying the reliability of tuning between two blocks of presentation of another stimuli (drifting grating; *Figure 5—figure supplement 2*). Notably, the tuning of units remained stable from one stimulus type to another: modulation of units during presentation of drifting gratings had a good correlation with their modulation during natural movie presentations for both blocks (*Figure 5—figure supplement 2d and h*). The tuning with running was even reliable between the first (30–90 min) and second (90–150 min) parts of the entire session, with each part containing different stimuli (*Figure 5—figure supplement 3*). We did a region-specific analysis of this reliability and found that reliable tuning exists in various regions (*Figure 5—figure supplement 3*). Overall, these analyses suggest that behaviour reliably and independently modulates neuronal responses.

## Stimulus dependence of behavioural variability and setpoint similarity

External stimulus directly modulates the responses by activating selective receptive fields of neurons, which can be measured under anaesthesia (*Niell and Stryker, 2008*; *Yoshida and Ohki, 2020*). In awake behaving animals, however, it is possible that different stimulus types indirectly modulate the responses by inducing different patterns of behavioural variability. We indeed found that this was the case when comparing natural movies with an unnatural stimulus (drifting gratings) (*Figure 6*). Natural movies induced more variability of pupil size and running in the animals across the two blocks of stimulus presentations: both measures significantly increased during the second block for natural movies, whereas changes were not significant for drifting gratings (*Figure 6a and d*). The result was consistent across the two datasets with different length and sequence of stimulus presentations (*Figure 5—figure supplement 3a and b*).

To see if and how this difference affects response similarity of units, we calculated average setpoint similarity (*Figure 4*) between the two blocks of presentations from the shuffled activity of units in response to different stimuli (see 'Methods'). Average setpoint similarity was high for both stimuli, but it was significantly larger for drifting gratings for most sessions (*Figure 6b and e*). Plotting setpoint similarity as a function of behavioural changes for the entire distribution revealed its composition across the two stimulus types. Responses to drifting gratings showed, on average, a higher setpoint similarity for similar behavioural states (small behavioural changes) (*Figure 6c and f*), arguing for more stability of average responses even independent of behavioural variability. Larger behavioural changes were more prevalent for the natural movie presentations, and units' responses showed a large drop of setpoint similarity at these deviations (*Figure 6c and f*), leading to a significant drop of average setpoint similarity compared to drifting gratings. Taken together, these results suggest that stability of population responses to different stimulus types might be determined by the combined effect of stimulus-evoked reliability of responses and its indirect modulation by behavioural variability.

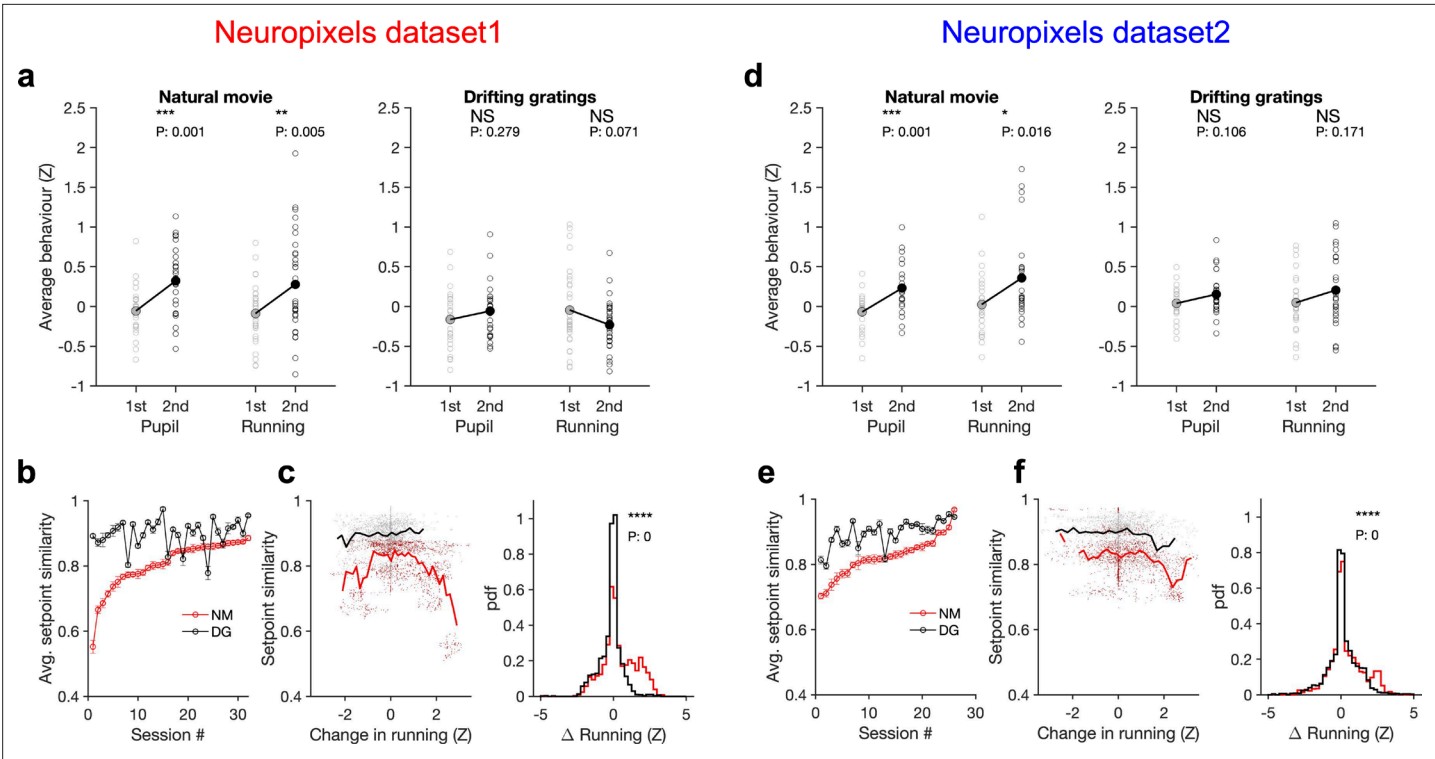

**Figure 6.** Stimulus dependence of behavioural variability and setpoint similarity. (**a**) Average pupil size and running speed during the first (grey) and second (black) blocks of presentation of natural movies (left) and drifting gratings (right) for different sessions (empty circles). Filled circles: the mean across sessions. Pupil size and running speed are z-scored across each session, respectively. p-Values on top show the result of two-sample *t*-tests between the two blocks. NS, p>0.05; *p≤0.05; **p≤0.01; ***p≤0.001; ****p≤0.0001. (**b**) Average setpoint similarity between the two blocks of presentation of natural movie 1 (NM) and drifting gratings (DG) for different sessions. Sessions are sorted according to their average setpoint similarity for NM. Population vectors are built out of the average responses of all units to 30 randomly chosen frames (1 s long). The correlation coefficient between a pair of population vectors from different blocks (within the same stimulus type) is taken as setpoint similarity. The procedure is repeated for 100 pairs in each session and the average value is plotted. Error bars show the SD across the repeats. (**c**), Left: setpoint similarity as a function of the difference in average running, $\Delta Z = Z_2 - Z_1$, where $Z_1$ and $Z_2$ are the average running during randomly chosen frames in the first and second block, respectively. The lines show the average of the points in 40 bins from the minimum $\Delta Z$ to the maximum. Right: distribution of changes in running for different stimuli. The probability density function (pdf) is normalized to the interval chosen (0.25). (**d–f**) Same as (**a–c**) for dataset 2. Source data (for the average z-scored pupil width and running speed in each block) are provided for individual sessions across the two datasets (***Figure 6—source data 1***).

The online version of this article includes the following source data for figure 6:

**Source data 1.** Related to ***Figure 6***.

## Decoding generalizability improves by focusing on reliable units

How does behavioural variability affect the decoding of stimulus-related information, and how can decoding strategies be optimized to circumvent the drift of representations? Our analyses so far suggested that behaviour modulates the responses in addition to and independently of stimulus-evoked modulations (independent model in ***Figure 2***). This independent behavioural modulation would be perceived as noise, if a downstream decoder is decoding stimulus-related signals, and can compromise the generalizability of decoding. For instance, the activity of a subpopulation of units (A) might be assigned to stimulus A by the decoder in the absence of significant behavioural modulation. If the decoder is now tested in a new condition where behaviour modulates the responses independently of the presented stimulus, the activity of subpopulation A can be interpreted as presence of stimulus A, independent of the presented stimulus. This is in contrast to the gain model (signal-gain model in ***Figure 2b***) in which behavioural state scales the stimulus-evoked signal, and can therefore not compromise the generalizability of decoding (subpopulation A only responds to stimulus A, but with different gains). In the signal-gain model, focusing on units which are strongly modulated by behaviour should in fact enhance the decoding generalizability under behavioural variability, whereas in the independent model the focus should be on units with more stimulus reliability.

We tested these two alternatives directly by training a linear decoder to discriminate between different frames of the natural movie (see *Figure 7a* and 'Methods'). The decoder was trained on the activity of units in the first block to detect a target frame; it was then tested on the second block of presentation to discriminate between the target frame and other frames in order to evaluate the generalizability of decoding (i.e. out-of-distribution transfer) (*Figure 7a*). When the decoder was trained on the activity of all units in the first block, discriminability ($d'$) was very low in the second block (*Figure 7b, c, e and f*). However, focusing on the reliable units (units with high stimulus reliability) shifted the distribution of $d'$ to larger values and increased the average discriminability (*Figure 7c and f*). Focusing on units with strong behavioural modulation, on the other hand, did not yield higher discriminability in the second block (*Figure 7—figure supplement 1*). These results suggest that behavioural modulation is detrimental to generalizability of stimulus decoding, and that this problem can be circumvented by focusing on units with more stimulus information.

This effect was consistent across mice in both datasets (*Figure 7d and g*). In dataset 2, we observed higher average $d'$ in the second block, when the decoder was trained and tested on all units. This could be due to more presentations of the natural movie in dataset 2 (30 repetitions in each block compared to 10 in dataset 1). Larger training samples can help the decoder in learning the signal from the noise, suggesting that the effect of behavioural 'noise' on corrupting the stimulus signal is more significant for small sample sizes. On the other hand, longer presentations can lead to sampling from responses under more behavioural variability, which can in turn inform the decoder on how to ignore the stimulus-irrelevant modulation by behaviour. Altogether, these results corroborate our previous analysis that the contribution of behavioural variability to neural activity is orthogonal to stimulus modulations and suggest that such behavioural noise limits the decoding capacity especially with limited data.

We also analysed how the behaviour can be decoded from the population activity (*Figure 7—figure supplement 2*). A decoder was trained on half of movie presentations (randomly chosen from the two blocks of presentation) to predict the pupil width and running speed, and was then tested on the other half of movie presentations (see 'Methods'). For the example sessions (shown in *Figure 1*), both behavioural parameters were predicted with high accuracy (*Figure 7—figure supplement 2a*). The accuracy was higher for dataset 2 (correlation of more than 90% for both parameters; compared to 60 and 80% for pupil width and running speed predictions, respectively, in dataset 1; *Figure 7—figure supplement 2a and b*). Similar results were obtained when the decoding of behavioural parameters was performed for other mice in both datasets (*Figure 7—figure supplement 2b*).

Higher prediction accuracy in dataset 2 is consistent with our reasoning above that longer episodes of stimulus presentation provide a higher chance for behavioural variability, which can in turn enable a decoder to learn the modulations arising from the behaviour better. An extreme case for this was when the decoder was trained on the first block in the example sessions and tested on the second block. For both datasets, the accuracy was very low (results not shown). This was a consequence of drastic changes of behaviour between the two blocks and specifically very little running in the first block (*Figure 1* and *Figure 5*); in such cases, the decoder cannot learn how behaviour modulates the neural activity from the training data due to paucity of relevant samples.

We also asked whether the decoding of behaviour may create any systematic bias in processing of sensory stimulus. If the behavioural modulation has a systematic relation with the modulation of neuronal activity by stimulus, a decoder which is predicting the animals' behaviour might be biased to infer illusionary stimuli due to the correlations between behaviour and stimulus. To test this, we weighted the responses of neurons to each movie presentation with the weights optimized for the behavioural decoder. We then calculated the correlation of the overall response of the decoder across multiple repeats of the natural movie. Systematic sensory bias in the read-out of the behavioural decoder should lead to significant correlations between the repeats of the same stimulus. Contrary to this, we observed correlations close to zero between different repeats, for the example sessions in both datasets, and across all mice (*Figure 7—figure supplement 2c*). These results suggest that behavioural decoding can be performed independent of sensory inference.

Taken together, the results of our behavioural decoding reveal two main insights: first, the behavioural state of the animal can reliably be predicted from the neural activity. This is consistent with our results on reliable modulation of neuronal activity by behaviour (*Figure 5*).

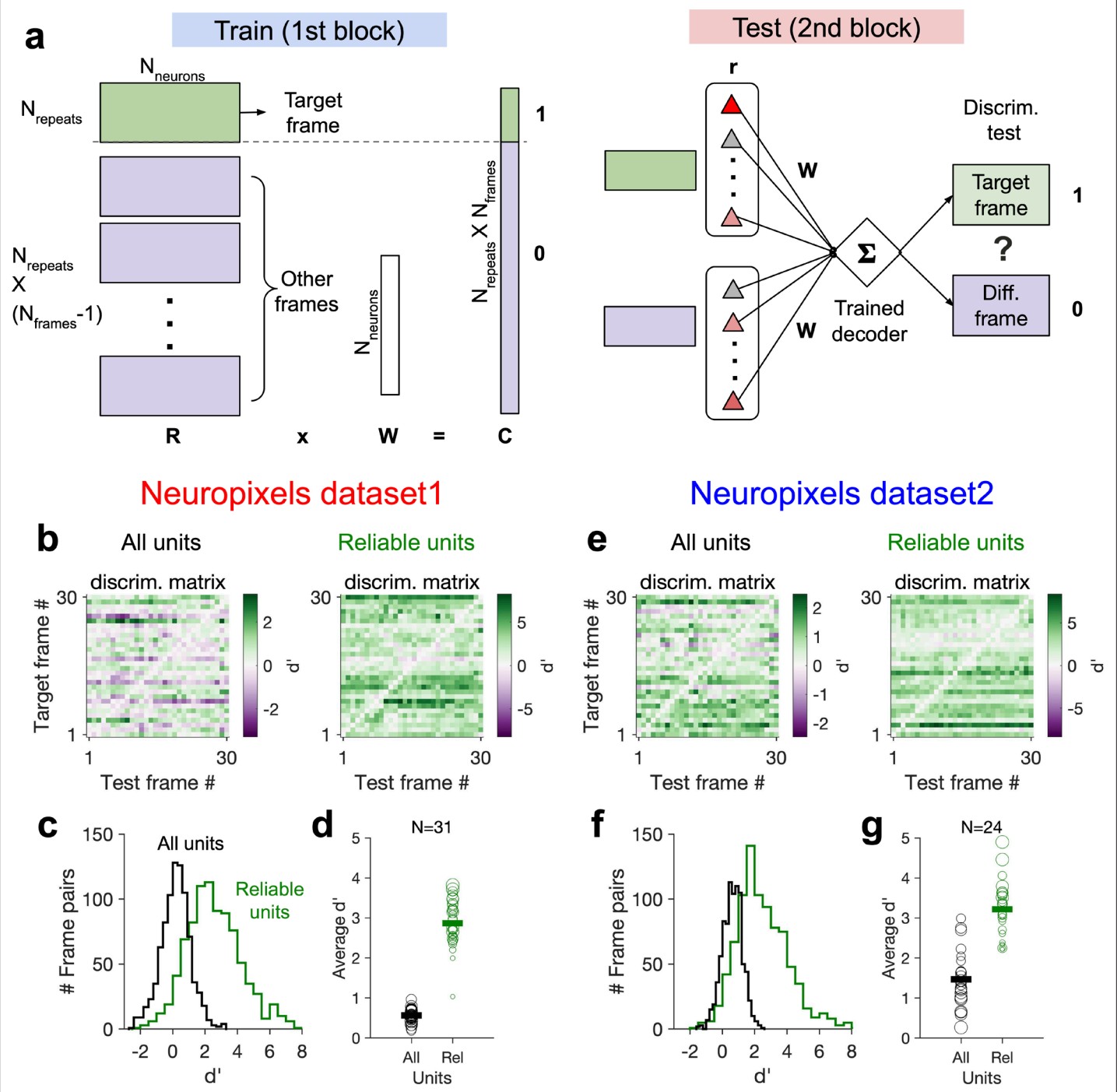

**Figure 7.** Decoding generalizability of natural images improves by focusing on reliable units. (**a**) Schematic of a decoder which is trained on the population activity during the first block of presentation of the natural movie (upper) and tested on the second block of presentation to discriminate different frames of natural images from each other (out-of-distribution transfer; see 'Methods' for details). (**b**) Matrix of discriminability index ('Methods'), *d'*, between all combinations of movie frames as target and test, when all units (left) or only units with high stimulus reliability (right) are included in training and testing of the decoder. (**c**) Distribution of *d'* from the example discriminability matrices shown in (**b**) for decoders based on all units (black) and reliable units (green). Reliable units are chosen as units with stimulus reliability ('Methods') of more than 0.5. (**d**) Average *d'* for all mice, when all units or only reliable units are included. Size of each circle is proportionate to the number of units available in each session (sessions with >10 reliable units are included). Filled markers: average across mice. (**e–g**) Same as (**b–d**) for dataset 2. Data in (**b, c**) and (**e, f**) are from the same example sessions shown in *Figure 1c* and *Figure 1f*, respectively.

The online version of this article includes the following figure supplement(s) for figure 7:

*Figure 7 continued on next page*

*Figure 7 continued*

**Figure supplement 1.** Decoding natural images does not improve by focusing on behaviourally modulated units.

**Figure supplement 2.** Decoding behavioural states from the population activity.

Second, behavioural decoding did not create a significant bias in sensory inference, which supports our previous results on independent modulation of neuronal responses by behaviour and sensory stimulus (*Figure 2* and *Figure 3*).

## Discussion

The results of our analysis here suggest that variability of the behavioural state of animal can contribute significantly to changes in representational similarity. We found that population responses to different repeats of the same natural movie were the most similar when behavioural parameters like pupil size and running speed changed the least. This was a result of an independent modulation of neural activity by behaviour, which was mixed with stimulus-evoked responses to represent a multidimensional code. Our results are consistent with a view in which behaviour modulates the low-dimensional, slowly changing setpoints of neurons, upon which faster operations like sensory processing are performed.

Small modulation of ongoing neural dynamics by sensory stimuli was reported before in awake, freely viewing animals (*Fiser et al., 2004*), in line with other reports on the significance of internal signals even in sensory cortices (*Arieli et al., 1996*; *Kenet et al., 2003*; *Tsodyks et al., 1999*). Our results here are consistent with these reports, and our analysis provides a mechanism by which variability of the internal state can contribute to ongoing signal correlations. It suggests that two distinct sources of response similarity exist in neuronal networks, with one set by baseline responses modulated on a slower timescale via internal parameters (setpoint similarity), and the other arising from finer and faster modulations invoked by sensory stimuli. Importantly, changes in representational similarity (which can lead to representational drift) can arise from changes in both sources, and hence attributing it purely to the drift of the sensory component might be inaccurate.

Internal, behavioural states of the animal can contribute independently to neural processing, or can act as a modulator for external stimuli, for instance, by increasing the input gain and enhancing the saliency of the sensory signal. Notably, our results could not be explained by a model in which behaviour acted as a gain controller for sensory inputs. Such a model would predict a direct relationship between the stimulus modulation and behavioural modulation of neurons. One would therefore expect that the most reliable neurons in representing sensory information to be modulated the most by arousal or running states. However, we found that the reliability of stimulus-evoked responses to different repeats of the same natural movie was independent of behavioural modulation, in line with a previous report (*Dipoppa et al., 2018*).

A gain-model account of behavioural modulation would only change the signal-to-noise ratio of sensory representations by behaviour. Therefore, if the level of arousal or attention of the animal drifts away over time, the signal component of the representations becomes weaker compared to the noise, leading to some drop in representational similarity. In contrast, independent modulation of neuronal responses by behaviour affects representational similarity in more complex ways. First, similarity of population vectors across repeats of the same stimuli can be due, at least partly, to the behavioural signal rather than stimulus-evoked responses. Second, changes in behavioural signal might be perceived as sensory-irrelevant noise if the parameter space of representations (composed of internal and external parameters) is only analysed over the external stimulus dimension (*Montijn et al., 2016*; *Stringer et al., 2019*). Reliable changes in behavioural signals might, therefore, be misinterpreted as the drift of stimulus-induced representations in the latter scenario.

Note that our results here do not rule out the contribution of other parameters, like slow latent learning and plasticity within the circuitry, leading to the drift of sensory representations, especially over longer timescales (days to weeks). Our analysis here revealed changes in neural representations in two blocks of multiple presentations of natural movies within the same day. To determine whether neuronal representations are gradually changing, there must be multiple (more than two) different compared time points (*Clopath et al., 2017*). Gradual change of neural representations over multiple time points can distinguish between representational drift and random neuronal variability which can arise independently in each repetition. In fact, we observed a gradual *increase* of both pupil width and

average activity of units during the entire session (*Figure 4—figure supplement 5*). Changes in the setpoint similarity arising from such gradual changes can therefore lead to representational drift over multiple time points. It would be interesting to repeat our analysis in future studies on other datasets which contain multiple blocks of stimulus presentations within and across days.

A recent analysis of similar datasets from the Allen Brain Observatory reported similar levels of representational drift within a day and over several days (*Deitch et al., 2021*). The study showed that tuning curve correlations between different repeats of the natural movies were much lower than population vector and ensemble rate correlations (*Deitch et al., 2021*) it would be interesting to see if, and to which extent, similarity of population vectors due to behavioural signal that we observed here (*Figure 4*) may contribute to this difference. In fact, previous studies showed gradual changes in the cells' activity rates during periods of spontaneous activity, suggesting that these changes can occur independently of the presented stimulus (*Deitch et al., 2021*; *Schoonover et al., 2021*).

The fact that there are changes that are not purely related to the tuning of cells is demonstrated in several previous studies. For instance, place cells in the mouse hippocampal CA1 can drop in and out of the ensemble during different visits to the same familiar environment over weeks, leading to a gradual change in the population of place cells that encode the location of the animal during the task (*Gonzalez et al., 2019*; *Ziv et al., 2013*). These changes, which reflect changes in activity rates, were independent of the position the neurons encode and were found in place cells and non-place cells alike. A similar turnover of the active cells in the cortex was also shown by *Aschauer et al., 2022* and *Driscoll et al., 2017*. Notably, *Rubin et al., 2015* showed that hippocampal representations of two distinct environments (which had different place-cell representations) co-evolve over timescales of days/weeks, with the shared component of the drift stemming from gradual context-independent changes in activity rates.

The stimulus-independent component of representational drift due to behavioural variability is a global phenomenon that can affect all regions, even independent of their involvement in the processing of natural images. In fact, we found similar representational drift in many areas, including regions like CA1 (see also *Deitch et al., 2021*), although units in this region had no reliable representation of natural images (*Figure 2—figure supplement 3a*). Global, low-dimensional modulation of activity by behavioural state, like arousal and running, or other aspects of behaviour that we did not have access to their quantification here (e.g. whisking, posture, or body movements other than running), might underlie such changes in representational similarity – although we cannot rule out the contribution of other factors like contextual modulations (as discussed above) or the passage of time (but see *Nilchian et al., 2022*), which might be more relevant to modulation of activity in regions like CA1. Drawing further conclusions about stimulus dependences of representational drift in visual cortex – and other sensory cortices – thus needs a critical evaluation by teasing apart the contribution of different components (stimulus-induced and stimulus-independent).

Another recent study reported stimulus-dependent representational drift in the visual cortex, whereby responses to natural images experienced large representational drift over weeks compared to responses to drifting gratings (*Marks and Goard, 2021*). In line with the finding of this study, we found here that responses to drifting gratings were more robust to behavioural variability in general. However, we also observed that different stimulus types can induce variable patterns of behaviour, thus highlighting the combined contribution of behaviour and stimulus to representational drift. Notably, the mentioned study *Marks and Goard, 2021* found a dependence of representational drift on the pupil size (see *Figure 2—figure supplement 3c* in *Marks and Goard, 2020*), with larger decreases in pupil size over time correlating with more representational drift for both stimulus types (see *Figure 4—figure supplement 1d* in *Marks and Goard, 2021*). Such consistent changes of behaviour may contribute to representational drift over longer timescales (days to weeks) by recruiting similar mechanisms as we described here for shorter intervals (e.g. changes in setpoint similarity). Mapping behavioural changes over longer times and per individual animal can shed light on the specific contribution of behaviour to representational drift. It would, for instance, be interesting to see if the large variability of representational drift across different animals (see *Figure 1—figure supplement 5* in the same study; *Marks and Goard, 2021*) might be linked to their behavioural variability.

Behavioural variability might be more pertinent to other modalities for which active sensing is less constrained during experiments. While eye movements are minimized in head-fixed mice, in other modalities (like olfaction) it might be more difficult to control for the behavioural variability arising

from active sensing (e.g. sniffing) over time and across animals. A recent study demonstrated significant representational drift over weeks in the primary olfactory cortex of mouse (*Schoonover et al., 2021*). The surprising finding that sensory representations are not stable in a sensory cortex was hypothesized to be linked to the different structure of piriform cortex compared to other sensory cortices with more structured connectivity. It would be interesting to see if, and to which extent, other factors like changes in the gating of olfactory bulb by variable top-down modulations (*Boyd et al., 2012*; *Markopoulos et al., 2012*), or changes in the sniffing patterns of animals, may contribute to this. Similar to the general decline over time of the pupil size reported in the visual cortex (*Marks and Goard, 2021*), animals may change their sniffing patterns during experiments, which can in turn lead to a general or specific suppression or amplification of odours, depending on the level of interest and engagement of individual animals in different sessions.

Behavioural modulation might be more systematically present depending on the design of specific tasks, for instance, if a sensory task is novel or otherwise engages behavioural states. Interpretation of the results of neural response to novel or surprising stimuli might, therefore, be compromised if one ascribes the changes in neural activity to local computations only, without the analysis of behaviour and without controlling for more global signals (e.g. arising from arousal, running, whisking, or licking). Low-dimensional signals associated with behavioural and internal state of the animal have in fact been suggested to create a potential confound for multisensory processing, with sound eliciting eye and body movements that modulate visual cortex, independent of projections from auditory cortex (*Bimbard et al., 2021*).

How can the animals perform reliable sensory processing in the face of extensive modulation of neural activity by behavioural changes? Our results suggest that the reliable units, which are stably encoding sensory information across behavioural states, could serve as a stable core for the brain to rely on for coping with changing neuronal responses (*Figure 7*). However, the distribution of stimulus reliability was surprisingly skewed towards lower reliability values (*Figure 2H*). In fact, at the population level, units were more reliably modulated by behaviour (*Figure 2H*). It is therefore likely that the brain would rely on both stimulus- and behaviour-related information, rather than focusing on a small subset of reliable units, to cope with changing representations.

Beyond sensory processing, variability of internal state can also contribute to other cognitive processes in various cortices (*Joshi and Gold, 2020*). A recent study in monkey found that changes in the perceptual behaviour were modulated by a slow drift in its internal state, as measured by pupil size (*Cowley et al., 2020*). This was correlated with a slow drift of activity in V4 and PFC, along with changes in the impulsivity of the animal (as reflected in the hit rates), which overrode the sensory evidence. These results, in another species, are in agreement with our findings here on the contribution of behavioural drift to changes in neural representations. Interestingly, the sensory bias model in the study could not capture the effect of the slow drift on decoding accuracy; instead, an alternative impulsivity model, which introduced the effect of slow drift as an independent behavioural parameter, matched with the data (Figure 6 in *Cowley et al., 2020*).

Another study in monkey M1 found that learning a new BCI task was modulated along the dimension of neural engagement of the population activity, which in turn was correlated with pupil size (*Hennig et al., 2021*). Neural engagement increased abruptly at the beginning, and decreased gradually over the course of learning, where output-null and output-potent components of neural engagement differentially attuned for different targets. Notably, exploiting behavioural perturbations in this study enabled an interactive interrogation of the neural code during learning. Behavioural perturbations, combined with large-scale recording and perturbation of neural activity (*Emiliani et al., 2015*; *Yizhar et al., 2011*; *Zhang et al., 2018*), which are more feasible in mice, can pave the way for a more precise (and potentially causal) interrogation of the neural mechanisms underlying representational drift. It would specifically be interesting to see how the bidirectional modulation of activity by behaviour we observed here emerges and which circuit mechanisms (*Cohen-Kashi Malina et al., 2021*; *Ferguson and Cardin, 2020*; *Fu et al., 2014*; *Pakan et al., 2016*) contribute to it.

In summary, our analysis reveals new insights on representational drift from the viewpoint of behaviour. Conceptually, it argues for the primacy of internal parameters (*Buzsáki, 2019*) and suggests that representational similarity could be better understood and characterized in a multidimensional parameter space where the contributions of both external and internal parameters are considered. Computationally, it argues for an independent mixing of stimulus-evoked and behavioural signals

rather than a simple gain modulation of sensory inputs by behaviour. Technically, it asks for further controls and analysis of behavioural variability in the characterisation of representational drift. Future studies will hopefully probe the multidimensional code underlying representations in the brain by combining large-scale recordings of neural activity with simultaneous measurement and quantification of behaviour.

## Methods
### Curation and preprocessing of the data
#### Data curation
Publicly available data provided by the Allen Brain Observatory (*de Vries et al., 2020*; *Siegle et al., 2021*) was accessed via AllenSDK (https://allensdk.readthedocs.io). We analysed recording sessions in which neuronal responses to visual stimuli were measured via electrophysiology techniques by Neuropixels probes (https://portal.brain-map.org/explore/circuits/visual-coding-neuropixels). The data composed of 58 sessions/mice in two separate datasets: brain observatory dataset (dataset 1; n = 32) and functional connectivity (dataset 2; n = 26) (*Supplementary file 1*). Similar stimuli (including natural moves and drifting gratings) were shown to the animals, with different length and sequence of presentations in each dataset (https://allensdk.readthedocs.io/en/latest/_static/neuropixels_stimulus_sets.png; see *Figure 5—figure supplement 3a and b* for illustration of different stimulus sets). We used the spiking activity of units which was already extracted by Kilosort2 (*Stringer et al., 2019*), and we included units in our analysis which passed the default quality criteria. Invalid intervals were treated as not a number (NaN) values. For further details on the preparation of animals, visual stimulation, data acquisition, and default preprocessing of data, see the Technical White Paper from the Allen Brain Observatory on Neuropixels Visual Coding.

#### Preprocessing of data
For our analysis here, we rendered the spiking activity of units in bins of 1 s. When analysis was focused on specific stimulus types (e.g. presentation of natural movie 1 as in *Figure 1d and g*), the activity was rendered from the onset of presentation of each block of the stimulus. When the analysis was across all stimuli and involved the activity during the whole session (e.g. data shown in *Figure 1c and f*), the activity was rendered from the beginning of the session or an arbitrary time (e.g. time frames specified in *Figure 5—figure supplement 3*). Behavioural information was obtained in similar time frames. Locomotion was quantified for all animals as the average running speed. Level of arousal was quantified by pupil size, as measured by pupil width (whenever pupillometry was available; *Supplementary file 1*).

To normalize the parameters (e.g. to normalize for different size of pupil across animals), we calculated their z-score values. For parameter $x$ (units' activity, pupil size, or running speed), it was obtained as $z = (x - \mu_x)/\sigma_x$, where $\mu_x$ and $\sigma_x$ are the mean and standard deviation of $x$ during the entire session or a specified time window.

### Data analysis
#### Representational similarity
Representational similarity of population activity was quantified by calculating the correlation of responses to different repeats of the same stimulus (*Figure 1—figure supplement 1c*). Let $v$ be a vector of responses of $N$ recorded units to $M$ 1-s-long chunks of a natural movie (the natural movie is broken down to $M$ chunks, or frames, each lasting for 1 s, corresponding to the bin width the neural activity is rendered in). $v$ is a $1 \times NM$ population vector composed of the concatenated activity of units (either the actual activity, i.e. average spiking activity, or the z-scored activity of each unit). Denote $v_i$ and $v_j$ as vectors of responses to two repeats of the same natural movie. Representational similarity is quantified as the Pearson correlation coefficient of these two population vectors:

$$\rho_{ij} = \frac{\text{cov}(v_i, v_j)}{\sigma_{v_i} \sigma_{v_j}}$$

RDI is calculated from representational similarity. Similar to previous metrics (*Marks and Goard, 2021*), we defined RDI between two blocks of stimulus presentation as

$$RDI = \left(CC_{ws} - CC_{bs}\right) / \left(CC_{ws} + CC_{bs}\right)$$

where $CC_{ws}$ and $CC_{bs}$ are, respectively, the average correlation coefficient of population vectors within and between sessions of presentation (see *Figure 1—figure supplement 5*). $CC_{bs}$ was obtained from the average of $\rho_{ij}$ for all pairs of repeats with $i$th repeat in the first block and the $j$th repeat in the second block of presentation. $CC_{ws}$ is obtained from the average of $\rho_{ij}$ for all non-identical pairs of repeats where the $i$th and $j$th repeats are both within the same block, respectively.

Note that RDI can also be defined, in principle, for a single pair of movie presentations (corresponding to blocks of presentation with 1 movie in each block). In this case, $CC_{ws}$ is 1, by definition, and we therefore have

$$rdi = \frac{1 - \rho_{ij}}{1 + \rho_{ij}}$$

showing the relation between representational drift and representational similarity in its most simplified case.

## Stimulus reliability

We also quantified the reliability of how single units respond individually to repetitions of the stimuli (*Figure 1—figure supplement 1d*). To quantify that, we calculated a stimulus reliability metric, which is obtained as the average correlation coefficient of each unit's activity vector ($r$) across repetitions of the stimulus (e.g. the natural movie). Let $r_i^k$ and $r_j^k$ be the vectors of response of the $k$th unit to the $i$th and $j$th repetitions of the natural movie. Similarity of the unit's response between these two repeats can be quantified by the Pearson correlation coefficient of the responses as before:

$$\rho_{ij}^k = \frac{\text{cov}\left(r_i^k, r_j^k\right)}{\sigma_{r_i^k} \sigma_{r_j^k}}$$

Stimulus reliability of the unit $k$ is obtained as the average correlation across all pairs of (non-identical) repetitions of the stimulus:

$$\rho^k = \frac{1}{N_r(N_r - 1)} \sum_{i=1}^{N_r} \sum_{j \neq i} \frac{\text{cov}\left(r_i^k, r_i^k\right)}{\sigma_{r_i^k} \sigma_{r_i^k}}$$

where $N_r$ is the number of repetitions of the stimulus. Note that to each single unit we can ascribe a stimulus reliability index since this is calculated from the individual vectors of single units' responses ($r^k$); on the other hand, representational similarity is calculated from the population vector of responses ($v$) and indicates a single population metric ascribed to the activity of a group of neurons (e.g. V1 units or all recorded units).

## Behavioural tuning

To obtain a measure of how single units are modulated by behaviour, we calculated the correlation of units' responses with behavioural parameter, $\beta$:

$$\rho_i\left(\beta\right) = \frac{\text{cov}\left(r_i, \beta\right)}{\sigma_{r_i} \sigma_\beta}$$

Here, $r_i$ is the vector of response of the $i$th unit, and $\beta$ is the vector of respective behavioural parameter (either pupil size or running speed) rendered during the same time window and with the same bin width as unit's activity.

To obtain a measure of reliability of this modulation by behaviour, we calculated bootstrap correlations. The activity of each unit was shuffled for 100 times and the correlation with behaviour was calculated. The mean ($\mu_{\text{sh}}$) and SD ($\sigma_{\text{sh}}$) of the distribution of shuffled correlations were then used to obtain the z-scored, bootstrapped correlation:

$$Z = \frac{\rho\left(\beta\right) - \mu_{\text{sh}}}{\sigma_{\text{sh}}}$$

where $\rho\left(\beta\right)$ is the unshuffled correlation of the unit with behaviour.

## Principal component analysis

To analyse the low-dimensional patterns of activity, we performed PC analysis on the vectors of population activity. The vectors of neuronal activity from which response similarity was calculated were of size $N \times N_r \times M$, where $N$ is the number of recorded units, $N_r$ is the number of repeats of the stimulus, and $M$ is the length of rendered activity in chunks of 1 s (corresponding to image frames of the natural movie). We concatenated these vectors to obtain a $N \times N_rM$, with each row denoting the total activity of each unit to all the repeats. PCs resulting from the PC analysis, therefore, represented a vector of length $N_rM$, denoting the low-dimensional activity of the population in the same time frame (each data point in *Figure 3b and h* corresponding to 1 s of activity). PC loadings (of size $N$ for each PC) are used to represent the individual units in the space of PCs (*Figure 3c, f, i and l*).

## Modelling

### Gain models

To gain mechanistic insight into the contribution of behavioural changes to modulation of representational similarity, we explored two different models. First, we developed a gain model, in which the integration of the signal and the noise by neurons was differently modulated by behaviour (*Figure 2—figure supplement 1a*). For a population of $N_p$ neurons, let $u$ be the $1 \times N_p$ vector of responses of neurons upon presentation of a stimulus. This is assumed to be composed of signal ($S$) and noise ($N$) components. Change in the behavioural parameters (for instance, pupil size) is supposed to change a gain parameter, $g$, which in turn differently modulate the signal ($S$) and noise ($N$). The vector of population activity, $u$, is obtained as a linear combination of weighted components by the behavioural/ gain parameter. If the signal and the noise are both scaled by the behavioural parameter, it is given as $u = gS + gN$. If either the noise or the signal is scaled, it is given as $u = S + gN$ or $u = gS + N$, respectively (*Figure 2—figure supplement 1b*).

$S$ and $N$ are both vectors of size $1 \times N_p$, where each element is drawn from a random uniform distribution between $[0, 1]$. The population activity is simulated for $N_r$ repeats of the stimulus. The stimulus signal, $S$, remains the same for all the repeats (frozen noise drawn from the same range as before, $[0, 1]$), while the noise component, $N$, is instantiated randomly on each repeat (from the same range, $[0, 1]$, as the signal). The behavioural parameter (e.g. pupil size) is assumed to change on every repeat too, which changes the gain parameter, $g$, as a result. $g$ was therefore assumed to be a random number uniformly drawn from the range $[0.5, 2]$ for each repeat. We chose $N_p = 1000$ and $N_r = 100$.

Representational similarity for different models was calculated, similar to the procedure in analysing the experimental data, as

$$\rho_{ij} = \frac{\mathrm{cov}\left(u_i, u_j\right)}{\sigma_{u_i} \sigma_{u_j}}$$

where $u_i$ and $u_j$ are population responses to the $i$th and $j$th repeat of the stimulus, obtained from different gain models. This value is plotted against the relative gain (obtained as the ratio of the gains in the two repeats, $g_i/g_j$ or $g_j/g_i$) in *Figure 2—figure supplement 1b*.

### Extended gain model

To match better with the experimental data on a single unit level, we extended the previous signal-gain model to have stimulus tuning for individual units (*Figure 2a*). Whereas before the stimulus was assumed to be a single, fixed value between $[0, 1]$ for each neuron, now the stimulus itself is extended (corresponding to different frames of the natural movie or different orientations of drifting gratings). The stimulus, $s$, is assumed to be a vector of fixed random values between [0,1] with size $1 \times N_s$. Each neuron, $k$, has a different stimulus drive/tuning, $T_k$, with which the stimulus vector is multiplied. $T$ is a vector of size $1 \times N_p$ (number of neurons in the population), randomly drawn from $[0, 1]$. Response of the $k$th neuron to each repeat of the stimulus is composed of its stimulus signal ($S = T_k s$), which is multiplied by the behavioural gain ($g$), and an added noise term ($N$), which is independently drawn for each stimulus and repeat from the range $[0, 1]$. $N_s = 10$, $N_p = 1000$, $N_r = 200$.

## Independent model

We also developed an alternative model, whereby the effect of behaviour on population responses was modelled as an independent signal (*Figure 2b*). Here, instead of scaling signal or noise components of the input, behaviour enters as an independent parameter:

$$u = S_S + N + S_B$$

where $S_S$ and $S_B$ are stimulus-evoked and behavioural signals and $N$ is the noise. $S_S$ and $N$ were instantiated as before, while $S_B$ was determined based on two factors. First, the behavioural parameter, $\beta$, which was changing on every repeat, and was simulated, similarly as the behavioural gains before, by a random number between $[0.5, 2]$ for each repeat. Second, the vector of tuning $(T_B)$ of different neurons in the population with the behavioural parameter, which was modelled as a random number between $[0, 1]$ for each neuron. The behavioural signal was obtained as $S_B = \beta T_B$. Representational similarity was computed as before for the population vectors and plotted against the relative behavioural parameters.

## Decoding model

To directly compare the stimulus-induced information available in different blocks of stimulus presentation, we developed a decoding model (*Figure 7a*). A linear decoder is trained on the neural activity (composed of the average activity of units in response to different repeats of the natural movie) during the first block of presentation to discriminate different frames (1 s long) of the natural movie (*Figure 7a*, upper). The weights of the readout (W) for each target frame were optimized to maximize its classification (C = 1) against other, non-target frames (C = 0). The decoder is then tested on the data in the second block (*Figure 7a*, lower). The population activity in response to each frame (the vector of average responses of neurons to a single frame across different repeats) is passed through the decoder to determine whether it is the target (D = 1) or not (D = 0). Performance of the decoder is quantified by calculating the discriminability (*d′*) as

$$d' = \frac{\mu_s - \mu_n}{\sqrt{\sigma_s^2 + \sigma_n^2}}$$

where $\mu_s$ and $\sigma_s$ are the average and SD of D for target frame across repetitions (within the second block), and $\mu_n$ and $\sigma_n$ are similar values for non-target frames. The discrimination matrix (*Figure 7b and e*) then shows the discriminability (*d′*) of each movie frame as a target when presented against all other frames.

## Theoretical analysis

### Gain models

Representational similarity for the responses in the gain models can be calculated as follows.

In the absence of any scaling of the signal or the noise, $u = S + N$, the representational similarity is obtained as the correlation coefficient of responses to a pair of stimulus repeats:

$$\rho_{ij} = \frac{\text{cov}(u_i, u_j)}{\sigma_{u_i} \sigma_{u_j}}$$

where $u_i = S + N_i$ and $u_j = S + N_j$. Assuming that $S$ and $N$ have zero means, we can write

$$\rho_{ij} = \frac{\sigma_S^2}{\sigma_S^2 + \sigma_N^2}$$

where $\sigma_S$ and $\sigma_N$ are the std of $S$ and $N$, respectively. This indicates that representational similarity can be expressed as a function of the relative variability of the signal and the noise. If modulation of the responses due to signal is dominant over the noise, $\sigma_S \gg \sigma_N$, $\rho_{ij} \rightarrow 1$.

If both the signal and the noise are scaled by the behavioural parameter, by the gain factor $g$, as $u = gS + gN$, we obtain:

$$\rho_{ij} = \frac{g_i g_j \sigma_S^2}{g_i g_j (\sigma_S^2 + \sigma_N^2)} = \frac{\sigma_S^2}{\sigma_S^2 + \sigma_N^2}$$

where $g_i$ and $g_j$ are the gains in the $i$th and $j$th repeat of the stimulus, respectively. Representational similarity, therefore, remains the same under similar scaling of $S$ and $N$.

If only the noise is scaled by behaviour, we obtain

$$\rho_{ij} = \frac{\sigma_S^2}{\sqrt{\left(\sigma_S^2 + g_i^2 \sigma_N^2\right)\left(\sigma_S^2 + g_j^2 \sigma_N^2\right)}}$$

showing that the larger the gain, the smaller the representational similarity.

Similarly, if only the signal is scaled, representational similarity can be obtained as follows:

$$\rho_{ij} = \frac{g_i g_j \sigma_S^2}{\sqrt{\left(g_i^2 \sigma_S^2 + \sigma_N^2\right)\left(g_j^2 \sigma_S^2 + \sigma_N^2\right)}}$$

which, if rewritten as

$$\rho_{ij} = \frac{\sigma_S^2}{\sqrt{\left(\sigma_S^2 + \sigma_N^2/g_i^2\right)\left(\sigma_S^2 + \sigma_N^2/g_j^2\right)}}$$

shows that larger gains effectively decrease the significance of noise, and hence enhance representational similarity. Specifically, in the limit of very large gains for both repetitions ($g_i \gg 1, g_j \gg 1$), we have $\rho_{ij} \to 1$.

For the specific case where gains are the same between the two repeats ($g_i = g_j = g$), the equation simplifies to

$$\rho_{ij} = \frac{\sigma_S^2}{\sigma_S^2 + \sigma_N^2/g^2}$$

Thus, for similar behavioural states (and hence gains) between the two repeats of the stimulus, representational similarity increases if $g > 1$ and decreases if $g < 1$.

## Independent model

For the model in which the stimulus and the behaviour contributes independently to neural responses, representational similarity in response to the same stimulus can be expressed as

$$\rho_{ij} = \frac{\sigma_S^2}{\sigma_S^2 + \sigma_B^2 + \sigma_N^2}$$

where $\sigma_S$, $\sigma_B$, and $\sigma_N$ denote the variability of the population response induced by stimulus, behaviour, and noise components, respectively. In deriving the above equation, we have assumed that the stimulus and behavioural components of the signal are independent, that is, $S_S.S_B \geq 0$ (in addition to the noise term being independent of $S_S$ and $S_B$, respectively). We also assumed that the behavioural signal, $S_B = \beta T$, remained the same between the two repeats (i.e. the behavioural parameter was the same: $\beta_i = \beta_j = \beta$). If the behavioural parameter changes between the repeats, the equation can, in turn, be written as

$$\rho_{ij} = \frac{\sigma_S^2}{\sqrt{\left(\sigma_S^2 + \beta_i^2 \sigma_T^2 + \sigma_N^2\right)\left(\sigma_S^2 + \beta_j^2 \sigma_T^2 + \sigma_N^2\right)}}$$

Note that, when representational similarity is only characterized in terms of the stimulus part of the signal, the contribution of behavioural variability is similar to a noise term – decreasing $\rho_{ij}$ for larger values of $\beta$. Changes in the behavioural state cannot, thus, be distinguished from random variability of the 'signal'.

## Relation between representational similarity and stimulus reliability

As explained above, representational similarity and stimulus reliability are calculated to quantify the similarity of population and single units' responses, respectively, to the repeats of the same stimulus. In fact, representational similarity of a population vector composed of one single unit is the same as the stimulus reliability of that unit. Similarly, if all the units in a population of neurons had the same

response profile in response to the stimulus, the stimulus reliability of units would be the same as the representational similarity of the population responses. Although these two measures are related (similar to lifetime sparseness and population sparseness *Froudarakis et al., 2014*), they are, however, not always directly equivalent to each other.

Consider a single unit, $k$, which has a constant baseline firing rate of $r_b$ and a component which is modulated by the stimulus, $r_m : r = r_b + r_m$ . If the stimulus-modulated component of the response is randomly changing between different repeats of the stimulus, the neuron would have a stimulus reliability of zero: $\rho^k = 0$. A population of units with this behaviour would have an average stimulus reliability of zero. However, the representational similarity of the responses of this population is not necessarily zero. In fact, we may obtain high values of population-level representational similarity, if the baseline component of the responses is significantly larger than their modulation ($r_b \gg r_m$). Under this scenario, representational similarity is calculated from the baseline component of the population responses ($v_b$), which indeed remains constant across repeats, hence $\rho_{ij} \to 1$.

## Acknowledgements

We thank KA Wilmes and DF Tome for their comments on the manuscript, and all the members of the Clopath Lab for insightful discussions. This work was supported by BBSRC BB/N013956/1, BB/N019008/1, Wellcome Trust 200790/Z/16/Z, Wellcome Trust 225412/Z/22/Z, Simons Foundation 564408 and EPSRC EP/R035806/1. All data needed to evaluate the conclusions in the paper are presented in the paper and/or the Supplementary Materials. Codes for reproducing the main analyses are provided.

## Additional information

### Funding

| Funder | Grant reference number | Author |
|---|---|---|
| Wellcome Trust | 225412/Z/22/Z | Sadra Sadeh |
| Wellcome Trust | 200790/Z/16/Z | Claudia Clopath |
| Biotechnology and Biological Sciences Research Council | BB/N013956/1 | Claudia Clopath |
| Biotechnology and Biological Sciences Research Council | BB/N019008/1 | Claudia Clopath |
| Simons Foundation | 564408 | Claudia Clopath |
| Engineering and Physical Sciences Research Council | EP/R035806/1 | Claudia Clopath |

The funders had no role in study design, data collection and interpretation, or the decision to submit the work for publication. For the purpose of Open Access, the authors have applied a CC BY public copyright license to any Author Accepted Manuscript version arising from this submission.

### Author contributions

Sadra Sadeh, Conceptualization, Data curation, Software, Validation, Investigation, Visualization, Methodology, Writing - original draft, Writing - review and editing; Claudia Clopath, Supervision, Funding acquisition, Writing - review and editing

### Author ORCIDs

Sadra Sadeh ⓘ http://orcid.org/0000-0001-8159-5461
Claudia Clopath ⓘ http://orcid.org/0000-0003-4507-8648

### Decision letter and Author response

Decision letter https://doi.org/10.7554/eLife.77907.sa1

Author response https://doi.org/10.7554/eLife.77907.sa2

## Additional files

### Supplementary files
- Supplementary file 1. Information of recording sessions in different datasets.
- Transparent reporting form
- Source code 1. Source Code for *Figures 1–4*.

### Data availability
All data needed to evaluate the conclusions in the paper are presented in the paper and/or the Supplementary Materials. Source Data Files have been provided for Figures 1, 2, 4 and 6 (uploaded as Excel files). Analysis code is uploaded as Source Code for Figures 1-4.

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
