## [Editor Report]

This work builds on rapidly accumulating evidence for the importance of measuring and accounting for behaviour in neural data, and will be of interest to a broad neuroscience audience. Analyses of Allen Brain Atlas datasets show that sensory representations change and match up reliably with behavioural state. The article's main conclusions are supported by the data and analyses, and the work raises important questions about previous accounts of the sources of representational drift in sensory areas of the brain.

---

## [Decision Letter]

**Decision letter after peer review:**

Thank you for submitting your article "Contribution of behavioural variability to representational drift" for consideration by *eLife*. Your article has been reviewed by 3 peer reviewers, one of whom is a member of our Board of Reviewing Editors, and the evaluation has been overseen by Michael Frank as the Senior Editor. The following individual involved in the review of your submission has agreed to reveal their identity: Yaniv Ziv (Reviewer #2).

Essential revisions:

The reviewers all enjoyed the paper and feel that the work is timely and well-executed. However, some common questions and concerns arose in the reviews which were discussed by the group. These are the essential revisions for the manuscript. Please also see the detailed comments from each reviewer below, which should improve the overall impact of the manuscript.

(1) Statistical analyses should not be limited to Pearson correlation and should include more sophisticated metrics, given the clear non-monotonic relationships in the data.

(2) Clarify the terms used that can seem at odds or confounding. Example: "representational drift" and "representational similarity" shouldn't be used interchangeably. For example, it may also be useful to provide working definitions of (and differences between) "drift of behavioral state", "behavioral drift" and "behavior" at the beginning of the manuscript.

(3) Related to point 2, it is crucial to explain the stance taken in the paper on how behavior and elapsed time are to be separated. Please see the relevant major prompts from Reviewers 2 and 3 who have suggested potential new analyses that would make this distinction most clear.

(4) The decoding results as currently reported seem obvious and underdeveloped. To make this have a bit more power, this should be expanded to include a new calculation, perhaps on decoding behavioral state along with the stimulus.

(5) Improve the figure layout to make the work more engaging to a broad readership. A diagram in Figure 1 would be useful.

(6) Show how these results depend on other behavioral parameters measured in these datasets, such as pupil location.

*Reviewer #1 (Recommendations for the authors):*

Overall the work is careful, and the presentation of the results is thorough. Some questions and suggestions for revision remain, however:

(1) The decoding results seem obvious – if the downstream decoder does not "know" about the behavioral state, then the neurons that are shifted by the behavioral state should be weighted less. Is there anything more meaningful or surprising that can be calculated here? What if the behavioral state is also read out from the population? Does the reliable shift in representation by behavior have any particular illusory effect, meaning does this create errors in a particular direction in visual space that could serve some functional purpose?

(2) What should one expect from this kind of analysis if a sensory task is novel or otherwise engages behavioral states more explicitly? Could the work be extended to comment on how to disentangle slow learning signals from the low-D modes that were found to be engaged by behavioral shifts?

(3) The figures, especially Figure 1, are dense and somewhat unintuitive. It would be nice to see a diagram of the analysis and main results first. This could engage a wider readership. Throughout the figures, it feels like there might be more visually impactful ways of representing the results. Instead, the figures feel repetitive, even though they are highlighting different effects and analyses.

*Reviewer #2 (Recommendations for the authors):*

1. Contribution of behavior vs. elapsed time: Given that it is already well-established that behavioral variability contributes to variability in neuronal responses (e.g. Niell and Stryker, Neuron 2010; Polack et al., Nat. Neurosci. 2013; Vinck et al., Neuron 2015; Musall et al., Nat. Neurosci. 2019; Stringer et al., Science 2019), it is not clear if the authors try to make the case that drift is an artifact of behavioral variability – i.e. drift is not a genuine phenomenon, it's all behavioral – as the abstract and main text strongly suggests by using the words "mistaken" and "misinterpreted"; or that behavior and time can both contribute to the observed changes in representations. These are two different messages that require a different set of analyses and will likely impact the field very differently, so it is crucial to clarify this point.

The authors state that "our results suggest that a significant fraction of what has been described as representational drift in a sensory cortex can be attributed to behavioral variability". However, quantifying the contribution of behavioral variability to representational drift (as the title of the paper claims) requires analysis that controls for the effects of the passage of time, and this was not done in this paper.

It is hard to conclude that both time and behavior modulation contribute differently to representational similarity of different presentations of the same video as the two analyzed video blocks are both well separated by time and differ in terms of the animal's behavior (quantified by pupil width and running speed). This will result in a correlation between time and behavioral changes that preclude the possibility to differentiate between the unique contribution of time and behavior to representational similarity.

One way to quantify the contribution of behavioral variability to representational similarity, is to repeat the analyses of Figure 1 and Figure 4, – i.e. the calculation of the correlation between the absolute change in the behavioral variables and the values of the chosen metric for representational similarity, while holding the interval between video repeats as a constant (subsequent video repeats within the same blocks or video repeats with the same absolute interval between them in general).

Alternatively, the authors can calculate and report the correlation between the change in behavior and representational similarity either between video repeats within or between video repeats across blocks. To control for the effects of time, the authors should separately calculate the linear relationship between the variables using only the light and dark gray data points (within block), and only the orange data points (between blocks), instead of the correlation indicated in red which uses the entire dataset. Additionally, the authors should report the statistics of these correlations (the p-value and the number of samples) in the corresponding figure legend.

The authors are suggesting that changes in representational similarity that were previously attributed to time are in fact only changes in the animal behavior over time. This could be tested using a multivariate regression analysis within each individual animal and quantifying the unique contribution of each variable (e.g., time, pupil size, running speed, etc.) as well as testing the significance of each variable to the fitted model, e.g. using GLM (see Driscroll et al., Cell 2017; Musall et al., Nat. Neurosci. 2019 for a similar approach).

Furthermore, if the authors are suggesting that representational drift merely reflects gradual changes in behavior, then it would be convincing to show that when the same analyses (e.g. multiple regression) are performed on a different set of stimuli with less abrupt changes in the behavioral state of the animals then there is no significant decrease in representational similarity. To do that, the authors can try to compare the similarity between the representations (for both population-vector correlations and setpoint similarity) across blocks in which the behavior didn't change significantly, and show there is no significant decrease in representational similarity. For example, the authors report in Figure 6 that there is less variability in behavior across blocks of drifting gratings and that there is no consistent and significant change in the behavior across the population of mice. Therefore, one way to test if there is higher representational similarity within blocks compared with across blocks (i.e., representational drift) without changes in behavior is to select mice that didn't exhibit behavioral changes across blocks and perform the analyses on them. This should also be done on the two blocks of 'Natural video 3', which are more proximal in time and therefore are likely to be more similar in behavior.

2. Cases of non-linear/monotonic relationship between behavioral changes and representational similarity and inappropriate use of correlation: In three of the main figures (specifically: Figure 1B, C, F, G, Figure 3C, I, and Figure 4C, D, G, H) the authors are using Pearson's correlation to quantify the relationship between the absolute change in pupil width between two video repeats and the representational similarity between the same repeats. In many of the plots, the data should not be fitted using Pearson's correlation since some of the assumptions of this model are not met. The most concerning issue is that the relationship between the two variables (change in behavior and representational similarity) does not always follow a linear or even a monotonic trend. This suggests that the relationship between the changes in pupil width and representational similarity are not correctly captured using a univariate linear model (e.g., Pearson's or Spearman's correlations). Additionally, in many of the plots, the data points fall into two or more dense clusters. This can lead to the false impression that there is a strong monotonic relationship between the two variables, even though there is a weak (or even opposite) relationship within each cluster (e.g., as in Figure 3 C, I and Figure 4G) (see an example in Aggarwal and Ranganathan, Perspect Clin Res 2016). This is a crucial point since the clusters of data points most likely represent different blocks that occurred at different times. Likewise, in Figure 5A, C, D, F right panels (and Extended Data Figure 10B, F), excluding the middle-range data points – which are the majority of data points – is unjustified and the use of Pearson's correlation in this case is inappropriate and misleading.

3. Stimulus dependent representational drift: A key statement of the current manuscript is that slow and gradual changes in behavior may change the setpoints (activity rates) of different neurons over time, leading to the appearance of a gradual decrease in representational similarity metrics such as 'population vector correlation' even in brain areas that do not, (literary) represent (or encode) the presented stimuli. This point was raised both as a critical evaluation of the representational similarity metrics chosen in the past to characterize the stability of visual representations or as a criticism of the use of the term 'representational drift'.

The fact that there are changes that are not purely related to the tuning of cells is not new and was demonstrated in several previous studies on coding stability. For instance, Ziv et al., Nature Neurosci. 2013 and Gonzales et al., Science 2019, have shown that place cells in the mouse hippocampal CA1 can drop in and out of the ensemble during different visits to the same familiar environment over weeks, leading to a gradual change in the population of place cells that encode the location of the animal during the task. These changes, which reflect changes in activity rates, were independent of the position the neurons encode and were found in place cells and non-place cells alike. Likewise, Driscoll et al., Cell 2017 and Aschauer et al., Cell Reports 2022, showed a similar turnover of the active cells in the cortex. Notably, Rubin et al., *eLife* 2015, showed that hippocampal representations of two distinct environments (which had very distinct place-cell representations) co-evolve over timescales of days-weeks. This shared component of the drift stems from gradual context-independent changes in activity rates. In fact, the prevalent use of the term representational drift (coined by Rule et al., Curr Op in Neurobiol 2019) is based on these above-mentioned studies and served to capture the entire range of gradual changes in neuronal activity over time. More recently, Schoonover et al., Nature 2021, and Deitch et al., Current Biology 2021 separately analyzed changes in activity rates and changes in tuning, and showed gradual changes in the cells' activity rates during periods of spontaneous activity, explicitly stating that these changes can occur independently of the presented stimulus.

Crucially, cells' activity rates are very relevant to stimulus encoding, as cells exhibit 'rate remapping' between environments or stimuli. Thus, equating stimulus-independent changes and changes in activity rates (setpoint similarity) is problematic because part of the changes in activity rates can be stimulus-dependent. This can be seen even in the visual cortex by calculating the setpoint similarity between different stimuli (e.g. Deitch et al., 2021 Figure S3H): the average activity rates of blocks of the same stimuli are more similar than between blocks of different stimuli. Thus, code stability is a function of two key factors: (1) stability in tuning and (2) stability in activity rates. Deitch et al., 2021 showed that the gradual changes over time that occur with respect to these factors are nearly independent of each other.

Furthermore, the fact that there are changes in setpoint similarity in CA1, although this area doesn't reliably encode visual stimuli, cannot, in itself, be used as an argument for the role of behavioral changes in representational drift since these changes can also be associated with elapsed time (see our point #1 above).

Overall, we agree that it is important to carefully dissociate between the effects of behavior on changes in neuronal activity that are stimulus-dependent or independent, but we feel that the criticism raised by the authors ignores the findings of the relevant literature, which (1) did not purely attribute the observed changes to the sensory component, and (2) did dissociate between stimulus-dependent changes (in tuning) and off-context/stimulus-independent changes (in activity rates).

We propose that the authors tone down their interpretations throughout the paper, and especially in the discussion: e.g., "changes in representational similarity (i.e. representational drift) can arise from changes in both sources, and hence attributing it purely to the drift of the sensory component might be inaccurate" and "Drawing further conclusions about stimulus-dependences of representational drift in visual cortex – and other sensory cortices – thus needs a critical evaluation by teasing apart the contribution of different components (stimulus-induced and stimulus-independent)".

4. The use of the term "Representational similarity" versus "Representational drift":

It is important that the authors explicitly define the term representational drift and edit the paper in a way that uses the terms "representational similarity" and "representational drift" in a consistent way throughout the manuscript. Most studies have demonstrated drift (even if not using the term 'drift') as a decreasing similarity between neuronal responses to the same stimulus/task as a function of the time interval between experiences/stimulus presentations under the same experimental conditions (Ziv et al., Nat. Neurosci. 2013; Lee et al., Cell 2020, Driscoll et al., Cell 2017; Rule et al., 2019; Schoonover et al., 2021; Deitch et al., 2021; Marks and Goard et al., Nat. Comm. 2021; Jensen et al., BioRxiv 2021 and Aschauer et al., Cell Reports 2022). This point regarding the differences between drift and variability in neuronal responses is nicely illustrated and discussed in a recent review paper (Clopath et al., Philos Trans, 2017). However, throughout the current manuscript, the authors refer to any change in representational similarity as representational drift and use these terms interchangeably regardless of the interval between the compared timepoints. For example:

"…Importantly, changes in representational similarity (i.e. representational drift)…"

In most of the above-mentioned studies about drift, the behavior or performance of the animal was at a steady state throughout the examined time intervals, suggesting that the observed changes in neuronal activity are not due to gradual changes in the behavior (e.g., due to learning, habituation, or changes in arousal). Thus, while the behavior itself may vary across different time points, as long as it is not changing gradually throughout the experiment, it should not lead to the appearance of drift.

To determine whether neuronal representations are gradually changing, there must be at least three (not necessarily equally spaced) different compared time points (see Clopath et al; 2017). We suggest adding a paragraph that explicitly explains the difference between neuronal variability and drift, how to differentiate between the two cases, and including an additional time point in the illustration presented in Extended Data Figure 1A (which now only includes two times points).

5. Focusing on reliable units improves time-lapse decoding: The analysis presented in Figure 7 shows that using reliable units (i.e., units that don't show changes in their tuning over time) results in higher decoding accuracy (i.e. more stable population code). Given that stability at the single-cell level should directly contribute to stability at the population level, this analysis is circular and therefore the conclusion that "Decoding generalizability of natural images improves by focusing on reliable units" is trivial.

Irrespective of this issue, we agree that it is a reasonable idea that reliable units could serve as a stable core for the brain to rely on for coping with changing neuronal responses. However, the distribution of stimulus reliability is not bi-modal (as shown in Figure 2H), but actually skewed towards lower reliability values. Thus, it is unclear how focusing on a small, unrepresentative subset of reliable units informs us how the brain copes with changing representations.

*Reviewer #3 (Recommendations for the authors):*

– In the introduction it may be useful to provide working definitions of (and differences between) "drift of behavioural state", "behavioural drift" and "behaviour".

– To rule out any potential artifact resulting from bin width choice being correlated with behavioral timescale, it would be useful to see the effect of varying bin width in computing population vectors.

– The Siegle et al. dataset methods imply that pupil position is available as well; thus do the same results apply if using position in addition to diameter? It would be nice to mention if any other behavioural measures are available that were not analyzed, as ignoring these seemed to lead previous accounts of drift astray.

– Pg. 5 bottom paragraph: "Inclusion of multiple cell types…to control for this…" – this control actually seems to be for possible strain differences; it is not clear in the Siegle at al. dataset how many cell types were presented (i.e. which cells were opto-tagged), so this information should be present if discussed.

– In the discussion of Figure 6, there is a nod to the expected results if pupil diameter correlation was used as a separate measure of behavioural tuning, which references Figure 6 b,e (pg. 13..should be Figure 5b,e?).

– Pg.19 "in line with previous reports" should be "in line with a previous report" unless more citations are provided.

– Pg. 20 "with a more decrease in pupil size" should be "with larger decreases in pupil size" or similar.

---

## [Author Response]

Essential revisions:The reviewers all enjoyed the paper and feel that the work is timely and well-executed. However, some common questions and concerns arose in the reviews which were discussed by the group. These are the essential revisions for the manuscript. Please also see the detailed comments from each reviewer below, which should improve the overall impact of the manuscript.(1) Statistical analyses should not be limited to Pearson correlation and should include more sophisticated metrics, given the clear non-monotonic relationships in the data.

That is an important point and we tried to address it by performing more sophisticated analyses and by developing new metrics, documented in multiple new figures (Figure 1—figure supplement 5; Figure 3—figure supplement 2; Figure 4—figure supplement 2; Figure 5). Specifically, we analyzed the source of the non-monotonic relationship (mentioned in Figure 4) in Figure 4—figure supplement 2 (see below for details). We also showed that the passage of time and the potential clustering arising from it is not a confound for our correlation analysis (see responses to point 3 below). In general, we took the approach of investigating the source of non-monotonicity by evaluating the contribution of different factors including various aspects of behavior and controlling for others like the passage of time. Importantly, by performing new analyses, we have now shown that the result of the linear correlation analysis is not compromised by the non-monotonic trend.

(2) Clarify the terms used that can seem at odds or confounding. Example: "representational drift" and "representational similarity" shouldn't be used interchangeably. For example, it may also be useful to provide working definitions of (and differences between) "drift of behavioral state", "behavioral drift" and "behavior" at the beginning of the manuscript.

Many thanks for this point. We clarified the definitions upfront, and explained the difference between different terms at the beginning of the Results section. We also clarified them in the Methods, in the figures and the captions*.*

(3) Related to point 2, it is crucial to explain the stance taken in the paper on how behavior and elapsed time are to be separated. Please see the relevant major prompts from Reviewers 2 and 3 who have suggested potential new analyses that would make this distinction most clear.

This is an important point. To elucidate that, we performed new analyses to show that elapsed time between the two blocks of presentations is not the major contributing factor to representational drift (Figure 1—figure supplement 5; see also Figure 3—figure supplement 2). More details are provided in response to the specific points raised by the reviewers below.

(4) The decoding results as currently reported seem obvious and underdeveloped. To make this have a bit more power, this should be expanded to include a new calculation, perhaps on decoding behavioral state along with the stimulus.

The suggestion of extending decoding to behavior is really interesting, and in fact it helped to further consolidate the main messages of the paper. To that end, we extended our decoding framework to predict behavior from neural activity (Figure 7—figure supplement 2). We found that, first, the behavioral state of the animal could reliably be predicted from the neural activity. This is consistent with our previous results on reliable modulation of neuronal activity by behavior (Figure 5). Second, we found that behavioral decoding did not create a significant bias in sensory inference, which supported our previous results on independent modulation of neuronal responses by behavior and sensory stimulus.

(5) Improve the figure layout to make the work more engaging to a broad readership. A diagram in Figure 1 would be useful.

We added diagrams at the beginning of Figure 1 to explain the main idea better, enhanced the illustration of the modelling figure (Figure 2), and generally tried to make other figures more engaging and intuitive.

(6) Show how these results depend on other behavioral parameters measured in these datasets, such as pupil location.

We analyzed and documented the dependence of setpoint similarity on pupil location (along with changes in average pupil location in each video presentation and the relationship of pupil location and pupil width) in two new figures (Figure 4—figure supplement 3,4). This information was also used, and proven to be helpful, in another analysis (see Figure 4—figure supplement 2).

Another parameter was pupil height, but this was highly correlated with pupil width, and we therefore decided to use one as a proxy for pupil size.

Reviewer #1 (Recommendations for the authors):Overall the work is careful, and the presentation of the results is thorough. Some questions and suggestions for revision remain, however:(1) The decoding results seem obvious – if the downstream decoder does not "know" about the behavioral state, then the neurons that are shifted by the behavioral state should be weighted less. Is there anything more meaningful or surprising that can be calculated here? What if the behavioral state is also read out from the population? Does the reliable shift in representation by behavior have any particular illusory effect, meaning does this create errors in a particular direction in visual space that could serve some functional purpose?

We agree that the decoding results were rather intuitive. The decoding framework was originally used to corroborate the main findings of our previous analyses (especially with regard to the relation between the modulation of units by stimulus and behavior), and to show how reliable sensory decoding can be possible in the face of behavioral variability.

The reviewer raises an interesting point as to how the decoding of behavior might work in tandem with sensory processing, and how this may affect sensory inference. To explore this question, we analyzed how different behavioral parameters (pupil width and running speed) can be decoded from the population activity (Figure 7—figure supplement 2).

First, we found that if the decoder is trained on stimulus representations that have enough behavioral variability, it can predict the behavior very well in the test sessions. This was consistent with our results on reliable modulation of neuronal activity by behavior. Second, we asked whether the decoding of behavior creates any systematic bias in sensory inference. If the behavioral modulation has a systematic relation with the modulation of neuronal activity with stimulus, a decoder which is predicting the animals’ behavior might be biased to infer illusionary stimuli due to the correlations between behavior and stimulus. To test this, we weighted the responses of neurons to each video presentation with the weights optimized for behavioral decoding. We then calculated the correlation of the readout response across multiple repeats of the natural video. We found very small correlations between repeats, indicating that behavioral decoding would not create illusionary cross-talks with the stimulus space. We think this is also consistent with our previous results on the independent modulation of activity by behavior.

(2) What should one expect from this kind of analysis if a sensory task is novel or otherwise engages behavioral states more explicitly? Could the work be extended to comment on how to disentangle slow learning signals from the low-D modes that were found to be engaged by behavioral shifts?

That is a very pertinent point. We believe it can specifically create confounds for tasks involving novelty as the animal would be engaged (in terms of attention or arousal) or even outwardly moving (in terms of running or whisking / licking) more during these stimuli. Interpretation of the results of neural response to novel or surprising stimuli might, therefore, be compromised, if one ascribes the changes (positive or negative) in neural activity to local computations, without the analysis of behavior and controlling for more global signals. Interestingly, another more recent dataset from the Allen Brain Institute has such behavioral tasks (https://portal.brain-map.org/explore/circuits/visual-behavior-2p) and it would be interesting to see what the analysis of neural activity and animals’ behavior reveals in this case. We added this to the Discussion.

(3) The figures, especially Figure 1, are dense and somewhat unintuitive. It would be nice to see a diagram of the analysis and main results first. This could engage a wider readership. Throughout the figures, it feels like there might be more visually impactful ways of representing the results. Instead, the figures feel repetitive, even though they are highlighting different effects and analyses.

Thank you for the feedback. We made Figure 1 simpler by removing more example sessions (moved to supplementary figures), and added illustrations of the main idea (in terms of analysis and results) at the beginning to communicate the gist upfront. We also tried to improve the other figures.

Reviewer #2 (Recommendations for the authors):1. Contribution of behavior vs. elapsed time: Given that it is already well-established that behavioral variability contributes to variability in neuronal responses (e.g. Niell and Stryker, Neuron 2010; Polack et al., Nat. Neurosci. 2013; Vinck et al., Neuron 2015; Musall et al., Nat. Neurosci. 2019; Stringer et al., Science 2019), it is not clear if the authors try to make the case that drift is an artifact of behavioral variability – i.e. drift is not a genuine phenomenon, it's all behavioral – as the abstract and main text strongly suggests by using the words "mistaken" and "misinterpreted"; or that behavior and time can both contribute to the observed changes in representations. These are two different messages that require a different set of analyses and will likely impact the field very differently, so it is crucial to clarify this point.

Sorry about the misunderstanding. We do not intend to claim that behavioral variability is the only source of drift. Drift of neuronal representations can arise due to a variety of sources, including synaptic turnover or synaptic plasticity, and via different mechanisms like feedforward, recurrent or top-down interactions (as we mention in the Introduction). Here, we are analyzing and clarifying the “contribution” of behavioral variability as one factor to this drift, and suggest that to reveal the contribution of other mechanisms, we need to control for the changes that can potentially arise from behavior (otherwise, those “real” effects might be mistaken or misinterpreted). We are sorry if this point was not clear in our previous writing, we tried to edit the text to reflect this message and tone better in the revised version. We also tried to clarify this by new analyses (Figure 1—figure supplement 5, and Figure 3—figure supplement 2).

The authors state that "our results suggest that a significant fraction of what has been described as representational drift in a sensory cortex can be attributed to behavioral variability". However, quantifying the contribution of behavioral variability to representational drift (as the title of the paper claims) requires analysis that controls for the effects of the passage of time, and this was not done in this paper.It is hard to conclude that both time and behavior modulation contribute differently to representational similarity of different presentations of the same video as the two analyzed video blocks are both well separated by time and differ in terms of the animal's behavior (quantified by pupil width and running speed). This will result in a correlation between time and behavioral changes that preclude the possibility to differentiate between the unique contribution of time and behavior to representational similarity.

This is an excellent point, we addressed it in the response above (also, see the new Figure 1—figure supplement 5, and the associated text and analyses.)

One way to quantify the contribution of behavioral variability to representational similarity, is to repeat the analyses of Figure 1 and Figure 4, – i.e. the calculation of the correlation between the absolute change in the behavioral variables and the values of the chosen metric for representational similarity, while holding the interval between video repeats as a constant (subsequent video repeats within the same blocks or video repeats with the same absolute interval between them in general).Alternatively, the authors can calculate and report the correlation between the change in behavior and representational similarity either between video repeats within or between video repeats across blocks. To control for the effects of time, the authors should separately calculate the linear relationship between the variables using only the light and dark gray data points (within block), and only the orange data points (between blocks), instead of the correlation indicated in red which uses the entire dataset. Additionally, the authors should report the statistics of these correlations (the p-value and the number of samples) in the corresponding figure legend.The authors are suggesting that changes in representational similarity that were previously attributed to time are in fact only changes in the animal behavior over time. This could be tested using a multivariate regression analysis within each individual animal and quantifying the unique contribution of each variable (e.g., time, pupil size, running speed, etc.) as well as testing the significance of each variable to the fitted model, e.g. using GLM (see Driscroll et al., Cell 2017; Musall et al., Nat. Neurosci. 2019 for a similar approach).

We agree that it would be ideal to have a GLM model to explain the contribution of each behavioral parameter and time to the neural activity. However, we did not find this approach easy or informative in this dataset. We think the reason is a complex, nonlinear and state-dependent modulation of neural activity by different aspects of behavior (see e.g. our new analysis in Figure 4—figure supplement 2), which hinders such a straightforward approach. We tried different approaches to address this issue, however, as we explain below.

Furthermore, if the authors are suggesting that representational drift merely reflects gradual changes in behavior, then it would be convincing to show that when the same analyses (e.g. multiple regression) are performed on a different set of stimuli with less abrupt changes in the behavioral state of the animals then there is no significant decrease in representational similarity. To do that, the authors can try to compare the similarity between the representations (for both population-vector correlations and setpoint similarity) across blocks in which the behavior didn't change significantly, and show there is no significant decrease in representational similarity. For example, the authors report in Figure 6 that there is less variability in behavior across blocks of drifting gratings and that there is no consistent and significant change in the behavior across the population of mice. Therefore, one way to test if there is higher representational similarity within blocks compared with across blocks (i.e., representational drift) without changes in behavior is to select mice that didn't exhibit behavioral changes across blocks and perform the analyses on them. This should also be done on the two blocks of 'Natural video 3', which are more proximal in time and therefore are likely to be more similar in behavior.

We performed such an analysis by quantifying the average behavioral change between the two blocks of presentation for individual mice and calculating a representational drift index (RDI) for each case (Figure 1—figure supplement 5). We found that strong representational drift mainly existed in those animals/sessions with large behavioral changes between the two blocks: there was in fact a strong correlation between average change in pupil width and RDI. Despite the passage of time, such drift was minimal in animals/sessions with small behavioral changes.

If the passage of time contributes mainly to changes in representational similarity between the two blocks, we should see comparable levels of representational drifts across animals with different levels of behavioural variability. Conversely, if changes in behavioural variability contributes mainly to changes in representational similarity between the two blocks, we should see stronger levels of representational drifts for animals with larger behavioural variability. We indeed found evidence in favour of the latter: Representational similarity remained rather stable for those animals which did not show large behavioural changes between the two blocks (Figure 1—figure supplement 5a). That is, passage of time per se did not contribute strongly to representational drift. Largest representational drifts were observed for animals with the largest changes in average pupil width between the two blocks (Figure 1—figure supplement 5a). In fact, there was a good correlation between the two across animals (Figure 1—figure supplement 5b).

The relationship was weaker in Neuropixels dataset2, whereby more repeats of the natural video are shown in each block (60 repeats for the total of 30 minutes, versus 20 repeats for 10 minutes in Neurpixels dataset1). Longer blocks of stimulus presentation increases the chance of behavioural changes within the blocks, which can in turn make the average block-wise representational similarity a less reliable metric. In line with this reasoning, further scrutiny into an outlier (with small average pupil changes but rather large representational drift between the two blocks; Figure 1—figure supplement 5b, right) revealed that changes in the running speed of the animal within each block can be the main contributing factor to changes in representational similarity (Figure 1—figure supplement 5c).”

2. Cases of non-linear/monotonic relationship between behavioral changes and representational similarity and inappropriate use of correlation: In three of the main figures (specifically: Figure 1B, C, F, G, Figure 3C, I, and Figure 4C, D, G, H) the authors are using Pearson's correlation to quantify the relationship between the absolute change in pupil width between two video repeats and the representational similarity between the same repeats. In many of the plots, the data should not be fitted using Pearson's correlation since some of the assumptions of this model are not met. The most concerning issue is that the relationship between the two variables (change in behavior and representational similarity) does not always follow a linear or even a monotonic trend. This suggests that the relationship between the changes in pupil width and representational similarity are not correctly captured using a univariate linear model (e.g., Pearson's or Spearman's correlations). Additionally, in many of the plots, the data points fall into two or more dense clusters. This can lead to the false impression that there is a strong monotonic relationship between the two variables, even though there is a weak (or even opposite) relationship within each cluster (e.g., as in Figure 3 C, I and Figure 4G) (see an example in Aggarwal and Ranganathan, Perspect Clin Res 2016). This is a crucial point since the clusters of data points most likely represent different blocks that occurred at different times. Likewise, in Figure 5A, C, D, F right panels (and Extended Data Figure 10B, F), excluding the middle-range data points – which are the majority of data points – is unjustified and the use of Pearson's correlation in this case is inappropriate and misleading.

We performed new analyses to shed light on the source of non-linear/non-monotonic relationships in our data. We show that this is arising as a result of more complex interactions between different aspects of behavior (Figure 4—figure supplement 2,3,4), and that it does not compromise our results when the separate clouds are excluded from the analysis (Figure 4—figure supplement 2).

In general, the opposite trend that exists in some sessions/animals can only compromise the linear regression analysis. The concern about the clustering arising from different blocks at different times is addressed separately (Figure 1—figure supplement 5, and Figure 3—figure supplement 2). It seems that the non-monotonic relationship is due to abrupt changes in different aspects of behavior (pupil width, location and running) and the way they modulate neuronal activity and representations. Precise analysis of this relationship needs more advanced multivariate and nonlinear / state-dependent models that can capture these complex interactions. But the relationship was not simple enough to be approached by simple addition of our univariate linear model to multivariate ones. Also, the nature of nonlinearity did not seem to be straightforward or the same across examples in the datasets, to be able to capture it by nonlinear correlation methods.

We want to emphasize that we did not use the linear regression to imply a linear relationship between the parameters, but to show and quantify the first-order dependence of representational (or setpoint) similarity on changes in behavior. More fine-tuned nuances and relationships should, of course, be further captured by more advanced analyses. We do not claim to have addressed the latter, as we believe such an analysis is beyond the scope of this work, which is mainly concerned with the first-order effect. By performing new analyses, we have now shown that the first-order relationship is neither compromised by the non-monotonic trends (Figure 4—figure supplement 2), nor by the concerns regarding the contribution of other factors like the passage of time to representational drift (Figure 1—figure supplement 5).

We hope to be able to expand our analysis in future studies to provide more explanatory and predictive power regarding the relationship between behavioral variability and neural activity. This is a hard problem, however, as the best models on the market can hardly capture more than 50% of variance (even in the absence of active behavior; see e.g. for a recent example: The Sensorium competition on predicting large-scale mouse primary visual cortex activity; https://arxiv.org/abs/2206.08666). But we hope that the insights obtained from our analysis here help in obtaining better models of encoding / decoding which are both informed by behavioral and sensory parameters.

With regard to quantification of behavioral tuning in Figure 5: The data points that are excluded (in Figure 5A, C, D, F right) are based on two criteria: bootstrap analysis (excluding units that do not show significant correlation with behavior), and comparable behavioral parameters between the two blocks (as shown in panels b,e). We are showing here that units that show significant modulation with running remain consistently modulated between the two blocks, provided that the animal has comparable level of running between the two blocks. Even if a unit would be modulated by behavior in principle, different levels of running could change the tuning (as we have discussed for extreme examples in our new decoding figure; Figure 7—figure supplement 2). Our results show that, focusing on such significantly modulated units and sessions, the units which are positively (/negatively) modulated by running in the first block remain positively (/negatively) modulated in the second block, and this trend is quantified by the linear regression.

Alternatively, we can quantify the number of units keeping their signs of modulation compared to those changing the sign between the two blocks, as a measure of constancy of tuning. We calculated a sign constancy index (SCI), which is the fraction of units preserving their sign of modulation. This quantification also led to similar results: when all units were included, we obtained a SCI of ~70% for both V1 and all units and for both datasets; when significantly modulated units in sessions with comparable levels of running were considered, SCI rose to more than 90% for both datasets, corroborating the results of our linear regression analysis.

3. Stimulus dependent representational drift: A key statement of the current manuscript is that slow and gradual changes in behavior may change the setpoints (activity rates) of different neurons over time, leading to the appearance of a gradual decrease in representational similarity metrics such as 'population vector correlation' even in brain areas that do not, (literary) represent (or encode) the presented stimuli. This point was raised both as a critical evaluation of the representational similarity metrics chosen in the past to characterize the stability of visual representations or as a criticism of the use of the term 'representational drift'.The fact that there are changes that are not purely related to the tuning of cells is not new and was demonstrated in several previous studies on coding stability. For instance, Ziv et al., Nature Neurosci. 2013 and Gonzales et al., Science 2019, have shown that place cells in the mouse hippocampal CA1 can drop in and out of the ensemble during different visits to the same familiar environment over weeks, leading to a gradual change in the population of place cells that encode the location of the animal during the task. These changes, which reflect changes in activity rates, were independent of the position the neurons encode and were found in place cells and non-place cells alike. Likewise, Driscoll et al., Cell 2017 and Aschauer et al., Cell Reports 2022, showed a similar turnover of the active cells in the cortex. Notably, Rubin et al., eLife 2015, showed that hippocampal representations of two distinct environments (which had very distinct place-cell representations) co-evolve over timescales of days-weeks. This shared component of the drift stems from gradual context-independent changes in activity rates. In fact, the prevalent use of the term representational drift (coined by Rule et al., Curr Op in Neurobiol 2019) is based on these above-mentioned studies and served to capture the entire range of gradual changes in neuronal activity over time. More recently, Schoonover et al., Nature 2021, and Deitch et al., Current Biology 2021 separately analyzed changes in activity rates and changes in tuning, and showed gradual changes in the cells' activity rates during periods of spontaneous activity, explicitly stating that these changes can occur independently of the presented stimulus.

We agree with that. We appreciate the hint into the previous literature, we added the reference to and discussion of these previous works to our Discussion.

Crucially, cells' activity rates are very relevant to stimulus encoding, as cells exhibit 'rate remapping' between environments or stimuli. Thus, equating stimulus-independent changes and changes in activity rates (setpoint similarity) is problematic because part of the changes in activity rates can be stimulus-dependent. This can be seen even in the visual cortex by calculating the setpoint similarity between different stimuli (e.g. Deitch et al., 2021 Figure S3H): the average activity rates of blocks of the same stimuli are more similar than between blocks of different stimuli. Thus, code stability is a function of two key factors: (1) stability in tuning and (2) stability in activity rates. Deitch et al., 2021 showed that the gradual changes over time that occur with respect to these factors are nearly independent of each other.

Thanks for raising this important point. We agree that the average activity can be different for different stimuli (e.g. as a result of different feedforward drive from full-filed gratings and natural images), that this can lead to different average setpoint similarity between repeats of stimulus types (as shown in Figure 6b,c,e,f), and that this can be very relevant for stimulus encoding. We did not intend to downplay the role of this difference. The aim was rather to highlight, within the same stimulus type, the difference between the average activity of a unit (independent of variations in stimulus – e.g. different frames of a natural video) and specific modulations arising from the change in the stimulus parameters (e.g. different image frames or orientations of gratings). We wanted to demonstrate that, within the same stimulus type, changes in the average activity independent of changes in the stimulus space can distort the measure of representational similarity that is often attributed to the variance in stimulus features (e.g. selectivity to natural images or preference to specific orientations). We agree that our language could be confusing, as the reviewer(s) mentioned, with regard to a more general picture of stimulus processing, which spans both mean and variance of stimulus-evoked responses, within and across different stimulus types. We therefore tried to refine our wording to avoid such confusion and clarify this distinction in the text now.

Furthermore, the fact that there are changes in setpoint similarity in CA1, although this area doesn't reliably encode visual stimuli, cannot, in itself, be used as an argument for the role of behavioral changes in representational drift since these changes can also be associated with elapsed time (see our point #1 above).

As we showed before, representational drift across animals/sessions was strongly determined by the degree of behavioral changes between the two blocks of presentation rather than passage of time (Figure 1—figure supplement 5). Such an effect is expected to be broadcasted to other regions too, and our analysis for all units show the same trend (Figure 1—figure supplement 5d,e). However, we cannot rule out the effect of other factors like the passage of time here. Note that passage of time in itself can be confounded by other (behavioral) changes. There might be changes in other aspects of behavior that we do not have access to their quantification (e.g. whisking, posture, or other body movements, as mentioned by the third reviewer below) and might be more relevant to regions like CA1. To err on the side of caution, we added a cautionary note to the discussion of our results now.

Overall, we agree that it is important to carefully dissociate between the effects of behavior on changes in neuronal activity that are stimulus-dependent or independent, but we feel that the criticism raised by the authors ignores the findings of the relevant literature, which (1) did not purely attribute the observed changes to the sensory component, and (2) did dissociate between stimulus-dependent changes (in tuning) and off-context/stimulus-independent changes (in activity rates).We propose that the authors tone down their interpretations throughout the paper, and especially in the discussion: e.g., "changes in representational similarity (i.e. representational drift) can arise from changes in both sources, and hence attributing it purely to the drift of the sensory component might be inaccurate" and "Drawing further conclusions about stimulus-dependences of representational drift in visual cortex – and other sensory cortices – thus needs a critical evaluation by teasing apart the contribution of different components (stimulus-induced and stimulus-independent)".

We edited our text and changed the language and tone to accommodate for these suggestions in order to appreciate the contribution of the average response, to refer to previous analyses, and to clarify what we mean in terms of the needed controls in future studies.

4. The use of the term "Representational similarity" versus "Representational drift":It is important that the authors explicitly define the term representational drift and edit the paper in a way that uses the terms "representational similarity" and "representational drift" in a consistent way throughout the manuscript. Most studies have demonstrated drift (even if not using the term 'drift') as a decreasing similarity between neuronal responses to the same stimulus/task as a function of the time interval between experiences/stimulus presentations under the same experimental conditions (Ziv et al., Nat. Neurosci. 2013; Lee et al., Cell 2020, Driscoll et al., Cell 2017; Rule et al., 2019; Schoonover et al., 2021; Deitch et al., 2021; Marks and Goard et al., Nat. Comm. 2021; Jensen et al., BioRxiv 2021 and Aschauer et al., Cell Reports 2022). This point regarding the differences between drift and variability in neuronal responses is nicely illustrated and discussed in a recent review paper (Clopath et al., Philos Trans, 2017). However, throughout the current manuscript, the authors refer to any change in representational similarity as representational drift and use these terms interchangeably regardless of the interval between the compared timepoints. For example:"…Importantly, changes in representational similarity (i.e. representational drift)…"

That is an important point and we tried to address it in the revised manuscript. Specifically, we clarified the definitions of the two terms and how we use them at the beginning.

We explain here why this confusion arises in our terminology, and how we tried to mitigate it.

First, we totally agree with the authors on the “conventional” definition of representational drift.

However, the datasets we were analyzing here were different in terms of timescale (hours) with the usual studies of drift (days to weeks), and we had access to limited repetitions of the stimuli. Nevertheless, the datasets are insightful and informative as they have recorded responses of large numbers of units in different areas to standardized stimuli across many animals while quantifying important (although yet incomplete) aspects of behavior.

We therefore focused on analyzing how representational similarity will be modulated by changes in behavior. As we explained before, any measure of representational drift (which tries to quantify the gradual changes over longer time scales) is eventually a function of representational similarity. We therefore based the main part of our analyses and results on describing how “changes in representational similarity” can be related to “behavioral changes”.

Once this is established, a consequence of our analysis would be that, if behavior gradually changes over the longer time course (for example, an animal loses motivation to be engaged in a task, or employs different strategies to sample the visual or olfactory stimulus space), this can lead to the representational drift (over the conventional time course).

Ideally, we wanted to have access to datasets with such a time course of repetition of stimuli and measurements of behavior. But in the absence of such standardized measurements, we used the current datasets to establish the dependence of “changes in representational similarity” on “behavioral changes” – which can in turn be the basis for “representational drift” in the conventional sense.

We, nevertheless, calculated a conventional representational drift index (RDI) between the blocks of presentation in the revised manuscript, and clarified our terminology to avoid confusion between those two terms.

In most of the above-mentioned studies about drift, the behavior or performance of the animal was at a steady state throughout the examined time intervals, suggesting that the observed changes in neuronal activity are not due to gradual changes in the behavior (e.g., due to learning, habituation, or changes in arousal). Thus, while the behavior itself may vary across different time points, as long as it is not changing gradually throughout the experiment, it should not lead to the appearance of drift.

We agree that many aspects of behavior might be at steady state over longer time courses.

But even over such long time courses, some gradual trends have been reported (e.g., gradual decrease of average pupil size or eye movement over weeks in the study of Marks and Goard, Nature Communications 2021: Supplementary Figure 11d and Supplementary Figure 10).

When active sensing is involved, it might be specifically difficult to control for changes in the behavioral strategy of the animal (e.g. active sampling of the visual or olfactory space).

Interestingly, a recent study has shown that accounting for different behavioral variability / strategies in bats reduces the apparent variability of the neural code (Liberti et al., Nature 2022). More studies with such datasets can shed light on the longer time scales of correlations and co-dependence between behavioral variability and the stability of neural code.

To determine whether neuronal representations are gradually changing, there must be at least three (not necessarily equally spaced) different compared time points (see Clopath et al; 2017). We suggest adding a paragraph that explicitly explains the difference between neuronal variability and drift, how to differentiate between the two cases, and including an additional time point in the illustration presented in Extended Data Figure 1A (which now only includes two times points).

We added a paragraph to our Discussion explaining the difference between neural variability and drift, commented on the gradual changes of the parameters, and added an additional time point in our illustration in Figure 1—figure supplement 1, as suggested by the reviewers.

The reviewers are completely right that establishing the drift of neural representations as a function of behavioral variability ultimately needs to be analyzed over longer time scales where a gradual drift of behavior is documented. However, as mentioned above, we did not have the needed data on this long time scale of recording (days to weeks) in the current datasets.

We therefore aimed to show how behavioral variability, even over this time scale, may affect the analysis of representational similarity (which would be the basis of representational drift), with the hope that the insights obtained from such analysis can be used in future for the datasets which measure large-scale population activity and quantify behavior over long time scales.

Within the same datasets, we now performed a new analysis to shed light on gradual changes (Figure 4—figure supplement 5). Our analysis shows that the average pupil size and the average activity of units are showing a concomitant gradual *increase* during the recording session – with a pattern remarkably similar between the two datasets (~50% increase in the average firing rate). A mechanism similar to what we analyzed in Figure 4 in terms of setpoint similarity can, therefore, contribute to gradual changes of representational similarity (and hence representational drift), for multiple repeats of stimulus presentations along this trajectory. It is also conceivable that similar gradual changes, e.g. *decrease* in pupil size over days and weeks (Marks and Goard, 2021), lead to similar effects.

5. Focusing on reliable units improves time-lapse decoding: The analysis presented in Figure 7 shows that using reliable units (i.e., units that don't show changes in their tuning over time) results in higher decoding accuracy (i.e. more stable population code). Given that stability at the single-cell level should directly contribute to stability at the population level, this analysis is circular and therefore the conclusion that "Decoding generalizability of natural images improves by focusing on reliable units" is trivial.Irrespective of this issue, we agree that it is a reasonable idea that reliable units could serve as a stable core for the brain to rely on for coping with changing neuronal responses. However, the distribution of stimulus reliability is not bi-modal (as shown in Figure 2H), but actually skewed towards lower reliability values. Thus, it is unclear how focusing on a small, unrepresentative subset of reliable units informs us how the brain copes with changing representations.

The main point of our (sensory) decoding analysis was to show the other side: that, in the face of strong behavioral variability, a decoder cannot generalize what it has learned from one block of stimulus presentation to another. We believe this is not an intuitive result in itself. We then continued to show that focusing on reliable units can rescue this and enable transfer of learning. We also used this decoding analysis to corroborate our previous results and analyses, on the independent modulation of neural responses by stimulus and behavior, and on reliable modulation of activity by behavior. We now also expanded our decoding analysis to decode behavior (Figure 7—figure supplement 2). These results might help to understand how the brain can cope with changing representations, if decoding of stimulus and behavior are both the targets. However, the questions about the reasons behind – and consequences of – strong modulation by behavior (and low reliability of stimulus responses) remain to be answered.

We speculated on the latter, along with the lines suggested by the reviewers, in a new paragraph in the Discussion.

Reviewer #3 (Recommendations for the authors):– In the introduction it may be useful to provide working definitions of (and differences between) "drift of behavioural state", "behavioural drift" and "behaviour".

This is an important point. We added these definitions and clarified their differences at the beginning of the text.

– To rule out any potential artifact resulting from bin width choice being correlated with behavioral timescale, it would be useful to see the effect of varying bin width in computing population vectors.

We extracted population vectors rendered in larger time bins and found similar dependence of representational similarity (calculated from these population vectors) with changes in pupil size (Figure 1—figure supplement 3b).

– The Siegle et al. dataset methods imply that pupil position is available as well; thus do the same results apply if using position in addition to diameter? It would be nice to mention if any other behavioural measures are available that were not analyzed, as ignoring these seemed to lead previous accounts of drift astray.

Many thanks for this great suggestion. We added the documentation of changes in eye position to the revised manuscript, and analyzed the dependence of changes in setpoint similarity on shift in the pupil center, and how this parameter was correlated with pupil width (Figure 4—figure supplement 3,4). This information was also helpful in other analyses we performed (Figure 4—figure supplement 2). The other behavioral parameter that is measured in the datasets and we did not include is pupil height. Since this was highly correlated with pupil width, we decided to use one as a proxy for pupil size.

– Pg. 5 bottom paragraph: "Inclusion of multiple cell types…to control for this…" – this control actually seems to be for possible strain differences; it is not clear in the Siegle at al. dataset how many cell types were presented (i.e. which cells were opto-tagged), so this information should be present if discussed.

That is correct. We have not analyzed the optotagging part of the experiments at the end. We have changed the wording to state that this control is for different transgenic mice, according to the information provided in Supplementary File 1.